# DIFFUSION-INSPIRED RECONFIGURATION OF TRANSFORMERS FOR UNCERTAINTY QUANTIFICATION

## ABSTRACT

Uncertainty quantification in pre-trained transformers is critical for their reliable deployment in risk-sensitive applications. Yet, most existing pre-trained transformers do not have a principled mechanism for uncertainty propagation through their feature transformation stack. In this work, we propose a diffusion-inspired reconfiguration of transformers in which each feature transformation block is modeled as a probabilistic mapping. Composing these probabilistic mappings reveals a probability path that mimics the structure of a diffusion process, transporting data mass from the input distribution to the pre-trained feature distribution. This probability path can then be recompiled on a diffusion process with a unified transition model to enable principled propagation of representation uncertainty throughout the pre-trained model's architecture while maintaining its original predictive performance. Empirical results across a variety of vision and language benchmarks demonstrate that our method achieves superior calibration and predictive accuracy compared to existing uncertainty-aware transformers.

## 1 INTRODUCTION

The transformer architecture (Vaswani et al., 2017) has become a universal backbone in most large-scale pre-trained or foundation models spanning numerous domains. These include language (Devlin et al., 2019; Radford et al., 2019; Brown et al., 2020; Achiam et al., 2023; Touvron et al., 2023), vision (Dosovitskiy et al., 2020; Touvron et al., 2021; Liu et al., 2021), speech (Baevski et al., 2020; Hsu et al., 2021; Radford et al., 2023), and even more complex domains with multi-modal data (e.g., text-image) (Radford et al., 2021; Liu et al., 2023; Driess et al., 2023; Team et al., 2023).

**Challenge.** Despite their prevalence, existing transformer-based models lack a principled mechanism to assess prediction uncertainty. This often leads to incorrect predictions being assigned high confidence (Guo et al., 2017; Mukhoti et al., 2020) which raises safety concerns in high-stake applications (Moon et al., 2020; Zhu et al., 2023) and underscores the importance of uncertainty quantification (UQ) in machine learning models. For example, effective UQ techniques can help determine when to defer to human experts in scenarios where the model exhibits high representation and/or prediction uncertainty, particularly in risk-sensitive applications (Tran et al., 2022a; Rudner et al., 2022b; 2023). While UQ has been extensively studied in conventional low-complexity deep neural networks, existing techniques mainly focus on imposing probabilistic priors on network weights and approximating their posteriors via either variational inference or posterior sampling. This quickly becomes both inaccurate and prohibitively expensive when the model complexity increases.

**Emerging Paradigm.** To sidestep the challenge of computing posteriors over models with exceedingly large complexities, there are emerging approaches that aim to reparameterize the attention outputs as (sparse) Gaussian process predictions (Liu et al., 2020; Chen & Li, 2023; Bui et al., 2025; Chen et al., 2024c) and recast the pre-trained transformer as a probabilistic chain mapping from the data distribution to a feature distribution. This enables principled, uncertainty-aware sampling of feature representations by simulating the chain rather than inferring them via computing the prohibitively expensive model posterior, thereby motivating a more scalable paradigm for UQ in large models.

**Research Gap.** Intuitively, separate reparameterization fails to account for the correlations among feature transformations at different attention blocks that were established during pre-training. This results in an unfavorable exchange between uncertainty calibration and performance. Previous approaches that adopt separate re-parameterization often improve uncertainty calibration at the cost

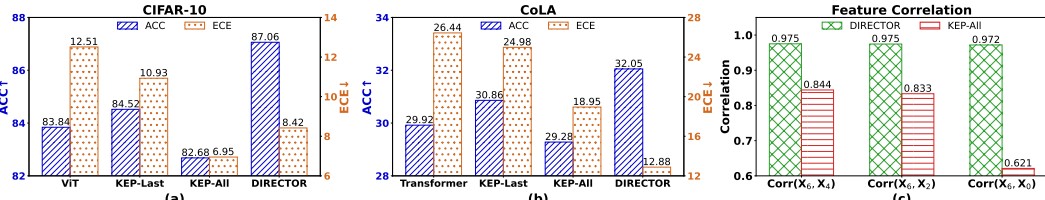

Figure 1: Comparison of accuracy (ACC↑) and uncertainty calibration (ECE↓) across pretrained models (ViT, Transformer), GP-reparameterized method KEP (Chen et al., 2024c) applied to either the last attention block (KEP-last) or all attention blocks (KEP-All), and our method (DIRECTOR) on (a) CIFAR-10 and (b) CoLA dataset. Panel (c) compares the correlation between features at the first layer ($\mathbf{X}_6$) and and those at deeper layers ($\mathbf{X}_4$, $\mathbf{X}_2$, and $\mathbf{X}_0$ at the last layer) for DIRECTOR and KEP-All on CIFAR-10 dataset.

of significantly reduced performance (see Fig. 1a, 1b), which does not align with the expectation that improved uncertainty calibration should, in general, lead to improved performance. From the above intuition, this is not surprising since accurate uncertainty propagation means being able to properly account for the high correlation across feature transformation steps. This is however not upheld in previous approaches according to our investigation. For instance, our results in Fig. 1c show that the correlation between the feature at the first KEP-SVGP's layer and intermediate representation rapidly decreases as it is propagated further towards the solution head. In contrast, our proposed method DIRECTOR manages to correctly preserve this high correlation. As a result, Fig. 1 shows that DIRECTOR indeed improves uncertainty calibration while also improving performance as expected.

**Solution Vision.** To address this gap, we propose distilling the sequence of reparameterized attention blocks into a unified diffusion model. Rather than treating each block as an independent reparameterization, we model the entire sequence as a continuous stochastic process over the feature embedding space. In this view, the observed transformations of a pre-trained model are interpreted as samples from a diffusion process governed by a single spatiotemporal transition kernel that maps the data distribution to the final representation. This unified view allows us to capture cross-block correlations established during pre-training while providing a principled mechanism for propagating uncertainty.

**Technical Contributions.** To substantiate this vision, we develop a diffusion-based framework for unified uncertainty propagation across transformer blocks with the following technical contributions:

**1.** We reinterpret the step-wise feature transformations of a pre-trained transformer as transition samples from a probabilistic path that maps the data distribution to the feature distribution. This perspective generalizes the transformer into a diffusion model parameterized by a unified transition kernel, which can be learned from these observed transitions. Such reconfiguration supports local uncertainty calibration at individual attention blocks while ensuring an accurate flow of uncertainty propagation across the entire network (see Section 2.1).

**2.** We design a training algorithm that distills the observed sequence of feature transformations into a unified spatiotemporal transition kernel of a diffusion process. The learned kernel captures the inherent correlations among feature transformations across attention blocks established during pre-training, providing a tractable and principled procedure for uncertainty quantification in large pre-trained models. This allows us to establish a generative paradigm for UQ, where uncertainty-aware representation samples are drawn directly from the learned diffusion process rather than inferred via intractable model posteriors (see Section 2.2).

**3.** We conduct extensive experiments on vision and language benchmarks to evaluate calibration quality, robustness, and out-of-distribution (OOD) detection. The results show that our approach consistently improves uncertainty quantification while preserving predictive performance over existing state-of-the-art pre-trained transformer models. Remarkably, it achieves these gains with fewer parameters than the original model, leading to improved memory efficiency. These results demonstrate the feasibility of post-hoc embedding probabilistic reasoning into the internal structure of large pre-trained models for uncertainty quantification without sacrificing performance. This opens a new direction for enhancing their reliability in safety-critical settings (see Section 3).

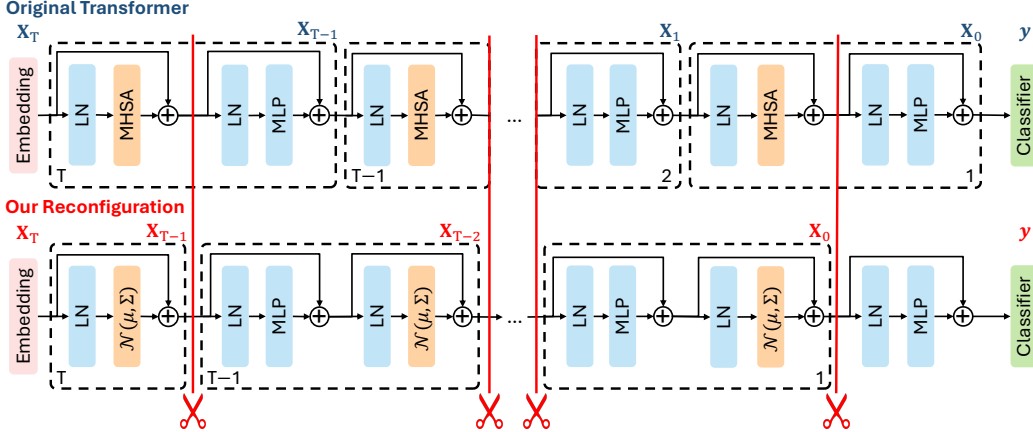

Figure 2: Restructuring a pre-trained transformer such that each block outputs a Gaussian distribution over its intermediate features, effectively aligning its architecture with a probabilistic path.

## 2    DIFFUSION-INSPIRED RECONFIGURATION OF TRANSFORMERS

Recent efforts to incorporate uncertainty into transformer-based models have uncovered a connection between multi-head self-attention (MHSA) and Gaussian processes (GPs) which shows that the deterministic output of a MHSA block corresponds to the posterior mean of a GP conditioned on its input (Chen & Li, 2023; Bui et al., 2025; Chen et al., 2024c) (see Appendix A.2). Although this insight offers a principled approach for uncertainty quantification at individual attention blocks, propagating it across MHSA blocks remain challenging. The difficulty arises because GP-reparameterized MHSA is interleaved with point-estimated components such as feed-forward networks (MLP) and layer normalization (LN). This disrupts the flow of uncertainty propagation since the interleaved sequence does not align with a well-defined stochastic path with a proper probabilistic transition model, particularly along the point-estimated segments of the pre-trained model.

To enable principled uncertainty propagation in pre-trained transformers, we instead propose a structural reconfiguration that reorganizes the model into a well-defined probabilistic path. In particular, we repartition the architecture such that each transformation block ends with an MHSA block (see Fig. 2), whose output can be interpreted as a Gaussian distribution over intermediate features. This restructuring instead views the aforementioned point-estimated network segments as additional parameterization of a GP-reparameterized MHSA rather than observations of a latent probabilistic transition (see Section 2.1). The resulting sequence of neuralized Gaussian transitions thus becomes well-aligned with the reverse-time stochastic process of a diffusion model.

These neuralized Gaussian transitions can then be viewed as discrete observations at different time steps of the diffusion's reverse-time process. We can thus learn this process via distilling these observed transition across different timesteps into a unified spatiotemporal transition kernel. This can be achieved via adopting variational inference as inspired by score-based diffusion methods (Sohl-Dickstein et al., 2015; Ho et al., 2020). This reveals a novel reconfiguration of pre-trained transformers into uncertainty-aware diffusion processes that interestingly enables UQ via learning generative models translating between raw data and predictive features (see Section 2.2).

### 2.1    RECONFIGURING PRE-TRAINED TRANSFORMER AS PROBABILITY PATH

Following prior work on uncertainty-aware transformers (Chen & Li, 2023; Bui et al., 2025; Chen et al., 2024c), the output of a kernelized attention head can be interpreted as the predictive mean at the input queries of a Gaussian process (GP) posterior conditioned on the key-value pairs. This means kernelizing attention transforms the original MHSA mechanism into a GP-based variant that naturally incorporates calibrated uncertainty. Each attention head thus admits a reparameterized Gaussian process (GP) structure which induces a neuralized Gaussian transition,

$$\boldsymbol{F}_t^{(h)} \mid \boldsymbol{U}_t \quad \sim \quad \mathbb{N}\left(\bar{m}_t^{(h)}(\boldsymbol{U}_t),\ \bar{\sigma}_t^{(h)}(\boldsymbol{U}_t)\right), \tag{1}$$

where $\boldsymbol{U}_t$ is the input of the $t$-th MHSA block whereas $\bar{m}_t^{(h)}(\boldsymbol{U}_t), \bar{\sigma}_t^{(h)}(\boldsymbol{U}_t)$ are neuralized mean and covariance functions for its $h$-th attention head under the reparameterization design (Appendix A.2).

The individual output representation $\boldsymbol{F}_t^{(h)}$ of each attention head $h$ of the $t$-th MHSA block can then be aggregated via a linear combination $\boldsymbol{O}_t$ which preserves the (neuralized) Gaussian structure:

$$\boldsymbol{R}_t \;=\; \boldsymbol{O}_t\left[\boldsymbol{F}_t^{(1)}, \boldsymbol{F}_t^{(2)}, \ldots, \boldsymbol{F}_t^{(n)}\right] \quad \Rightarrow \quad \boldsymbol{R}_t \mid \boldsymbol{U}_t \;\sim\; \mathbb{N}\!\left(\bar{m}_t(\boldsymbol{U}_t), \bar{\sigma}_t(\boldsymbol{U}_t)\right), \qquad (2)$$

where $\bar{m}_t(\boldsymbol{U}_t) \triangleq [\boldsymbol{O}_t\bar{m}_t^{(1)}(\boldsymbol{U}_t), \ldots, \boldsymbol{O}_t\bar{m}_t^{(n)}(\boldsymbol{U}_t)]$ and $\bar{\sigma}_t(\boldsymbol{U}_t) \triangleq \text{blkdiag}[\boldsymbol{O}_t\bar{\sigma}_t^{(h)}(\boldsymbol{U}_t)\boldsymbol{O}_t^\top]$.

This reparameterization thus reconfigures a pre-trained (point-estimate) MHSA block into probabilistic transition function with a uncertainty structure which can be optimized as in previous methods (Chen & Li, 2023; Bui et al., 2025; Chen et al., 2024c). This provides a principled handle for uncertainty calibration. One can assess the local representation uncertainty via the (learned) predictive variance or generate uncertainty-aware representation samples to propagate downstream. This propagation is however disrupted in existing approaches as mentioned previously due to the interleaving of MHSA with point-estimated components such as MLPs and layer normalization (LN). To elaborate, the transformation from one intermediate representation $\boldsymbol{X}_{t-1}$ to the next $\boldsymbol{X}_t$ interleaves the MHSA mechanism with the MLP and LN mechanisms:

$$\boldsymbol{X}_{t-1} \;=\; \text{MLP}(\text{LN}(\boldsymbol{Z}_t)) + \boldsymbol{Z}_t \quad \text{where} \quad \boldsymbol{Z}_t \;=\; \text{MHSA}\left(\text{LN}(\boldsymbol{X}_t)\right) + \boldsymbol{X}_t\,, \qquad (3)$$

where we number transformer block in reverse such that the first transformer block is indexed with $t = T$ and the last is indexed with $t = 0$, as illustrated in the upper part of Fig. 2.

To propagate uncertainty under this partition, the point-estimated network segment $\text{MLP}(\text{LN}(\boldsymbol{Z}_t))$ can be viewed as an observed function sampled from some function prior. However, unlike the MHSA which can be viewed as a sampled function from a learnable Gaussian process (GP) prior as established in prior works, it remains unclear how to parameterize and learn a function prior for $\text{MLP}(\text{LN}(\boldsymbol{Z}_t))$ without the risk of prior misspecification. Otherwise, treating it as a deterministic transition collapses the uncertainty structure and consequently disrupts uncertainty propagation.

To sidestep this technical challenge, we propose to instead view $\text{MLP}(\text{LN}(\boldsymbol{Z}_t))$ as an additional parameterization of the neuralized Gaussian transition induced by the GP-reparameterized MHSA. This can be achieved via a rearrangement of transformer's computation blocks as detailed below:

$$\boldsymbol{X}_{t-1} \;=\; \text{MHSA}(\text{LN}(\boldsymbol{Z}_t)) \;+\; \boldsymbol{Z}_t \text{ where } \boldsymbol{Z}_t \;=\; \begin{cases} \text{MLP}\left(\text{LN}(\boldsymbol{X}_t)\right) + \boldsymbol{X}_t, & \text{if } t \neq T \\ \boldsymbol{X}_T, & \text{otherwise} \end{cases}. \quad (4)$$

This reconfiguration guarantees that each re-partitioned computation block terminates with an MHSA module (see the lower part of Fig. 2). The deterministic transition $\text{MLP}(\text{LN}(\boldsymbol{Z}_t))$ now become additional parameters of the MHSA which can be reparameterized into a neuralized Gaussian transition. Note that the skip connection does not break Gaussianity but only shifts the mean. Consequently, this construction induces a stochastic process $\{\mathbf{X}_t\}_{t=0}^T$ with Gaussian transitions:

$$p\!\left(\boldsymbol{X}_{t-1} \mid \boldsymbol{X}_t\right) \;=\; \mathbb{N}\!\left(\boldsymbol{X}_{t-1} \mid m_t(\boldsymbol{X}_t),\; \sigma_t(\boldsymbol{X}_t)\right), \qquad (5)$$

where $m_t(\boldsymbol{X}_t) = \bar{m}_t(\text{LN}(\boldsymbol{Z}_t)) + \boldsymbol{Z}_t$ and $\sigma_t(\boldsymbol{X}_t) = \bar{\sigma}_t(\text{LN}(\boldsymbol{Z}_t))$ with $\boldsymbol{Z}_t$ is defined in Eq. 4. These separately parameterized Gaussian transitions across timesteps can be distilled into a unified spatiotemporal Gaussian transition that defines the reverse-time process of a diffusion model as discussed in Section 2.2. This unified parameterization enables seamless uncertainty propagation while explicitly encoding transition correlations across steps as desired.

**Remark.** We note that the above reconfiguration does not alter the pre-trained computation but it does change how the point-estimated segment $\text{MLP}(\text{LN}(\boldsymbol{Z}_t))$ is interpreted. Rather than being an observed function drawn from an unknown prior, it is parameterized as part of a Gaussian transition. This reveals a learnable representation medium that is more amenable to uncertainty propagation. For ease of presentation, we also abuse the notation $\boldsymbol{X}/\boldsymbol{F}$ to denote $\text{vec}(\boldsymbol{X})/\text{vec}(\boldsymbol{F})$.

## 2.2 DISTILLING TRANSFORMER-BASED PROBABILITY PATH ON DIFFUSION MODEL

Under our proposed reconfiguration in Section 2.1, the transformer induces a stochastic path $\{\mathbf{X}_t\}_{t=0}^T$ with Gaussian transitions (Eq. 5), which closely resembles a reverse diffusion process. This defines a

distribution over intermediate features conditioned on the original input embedding $\boldsymbol{X}_T$:

$$p\big(\mathbf{X}_{T-1},\ldots,\mathbf{X}_0 \mid \boldsymbol{X}_T\big) \;=\; \prod_{t=1}^{T} p\big(\mathbf{X}_{t-1} \mid \mathbf{X}_t\big) \;=\; \prod_{t=1}^{T} \mathbb{N}\Big(\boldsymbol{X}_{t-1} \mid m_t(\boldsymbol{X}_t),\, \sigma_t(\boldsymbol{X}_t)\Big)\,, \tag{6}$$

where the transition probability $p(\mathbf{X}_{t-1} \mid \mathbf{X}_t)$ is modeled independently at each timestep. Calibrating uncertainty in this block-wise, decoupled structure is difficult as it does not capture transition correlation and hence does not generalize across timesteps.

To address this limitation, we require a more parsimonious representation that characterizes the entire sequence of transition models in a unified manner. This can be achieved by re-compiling it into a reverse-time diffusion process with a unified spatiotemporal transition model:

$$q_\theta\big(\mathbf{X}_{T-1},\ldots,\mathbf{X}_0 \mid \boldsymbol{X}_T\big) \;=\; \prod_{t=1}^{T} q_\theta\big(\mathbf{X}_{t-1} \mid \mathbf{X}_t\big) \;=\; \prod_{t=1}^{T} \mathbb{N}\Big(\boldsymbol{X}_{t-1} \mid m_\theta(\boldsymbol{X}_t),\, \sigma_\theta(\boldsymbol{X}_t)\Big)\,, \tag{7}$$

In particular, the entire sequence of neuralized Gaussian transitions derived from the previously described GP-reparameterized of pre-trained transformer can be absorbed into the reverse-time diffusion with a unified spatiotemporal transition via minimizing the following negative log-likelihood, analogous to score matching in diffusion models (Sohl-Dickstein et al., 2015; Ho et al., 2020):

$$L(\theta) \;=\; \mathbb{E}_{p(\boldsymbol{X}_0 \mid \boldsymbol{X}_T)}\Big[ -\log q_\theta\big(\boldsymbol{X}_0 \mid \boldsymbol{X}_T\big)\Big]\,. \tag{8}$$

This negative log-likelihood (NLL) loss admits the following upper-bound via variational inference:

$$L(\theta) \;\leq\; \mathbb{H}\Big(p(\mathbf{X}_0 \mid \mathbf{X}_T)\Big) + \sum_{t=1}^{T} \mathbb{E}_{p(\mathbf{X}_t \mid \mathbf{X}_T)}\Big[ D_{\mathrm{KL}}\Big(p(\mathbf{X}_{t-1} \mid \mathbf{X}_t) \,\|\, q_\theta(\mathbf{X}_{t-1} \mid \mathbf{X}_t)\Big)\Big]\,, \tag{9}$$

with proof deferred to Appendix A.4. Since the entropy term $\mathbb{H}\big(p(\mathbf{X}_0 \mid \mathbf{X}_T)\big)$ is independent of $\theta$, optimizing the bound in Eq. 9 reduces to minimizing the Kullback-Leibler (KL) divergence:

$$L_1(\theta) \;=\; \mathop{\mathbb{E}}_{t \,\sim\, \mathbb{U}(1,T)} \mathop{\mathbb{E}}_{\boldsymbol{X}_t \sim p(\mathbf{X}_t \mid \mathbf{X}_T)}\Big[ D_{\mathrm{KL}}\Big(p(\boldsymbol{X}_{t-1} \mid \boldsymbol{X}_t) \,\|\, q_\theta(\boldsymbol{X}_{t-1} \mid \boldsymbol{X}_t)\Big)\Big]\,, \tag{10}$$

where $t \sim \mathbb{U}(1,T)$ and $\boldsymbol{X}_t$ is sampled via sampling data $\boldsymbol{X}$ and simulating the corresponding output of the $(T-t)$-th block of the pre-trained transformer. This loss aligns the probability path with a diffusion-style transition kernel while enabling generalization across timesteps. To ensure that the learned uncertainty propagation process maps from data to feature distributions which are informative for downstream prediction, we regularize it with an additional performance loss:

$$L_2(\theta) \;=\; \mathop{\mathbb{E}}_{(\boldsymbol{X},\boldsymbol{y})\sim \boldsymbol{D}} \mathop{\mathbb{E}}_{\boldsymbol{X}_0 \sim q_\theta(\boldsymbol{X}_0 \mid \boldsymbol{X}_T)}\Big[ \mathrm{loss}\Big(\boldsymbol{X}_0, \boldsymbol{y}\Big)\Big]\,, \tag{11}$$

where $\boldsymbol{X}$ is sampled from the training dataset $\boldsymbol{D}$ and is embedded with $\boldsymbol{X}_T = \mathrm{embed}(\boldsymbol{X})$. $\boldsymbol{X}_0$ is then sampled via iteratively simulating the current estimate of the probability path $q_\theta(\boldsymbol{X}_{t-1} \mid \boldsymbol{X}_t)$. The parameterization of the unified spatiotemporal transition model can then be obtained via:

$$\theta \;=\; \arg\min_\theta \;\Big\{L_1(\theta) \;+\; L_2(\theta)\Big\}\,, \tag{12}$$

which combines the uncertainty-aware (reconfiguration) loss with the performance-aware loss. For implementation details of the above algorithm, please refer to Appendix A.5.

## 3 EXPERIMENTS

This section evaluates the efficacy of our proposed method, DIRECTOR: **D**iffusion-**I**nspired **REC**onfiguration of **T**ransf**OR**mers for Uncertainty Quantification, by reconfiguring existing uncertainty-aware transformers into diffusion-based models and comparing their uncertainty calibration and predictive performance against those of the original versions. We describe our experiment settings in Section 3.1 and report detailed empirical results in Section 3.2.

## 3.1 EXPERIMENT SETTINGS

**Datasets.** We evaluate DIRECTOR using datasets in computer vision (CV) and natural language processing (NLP). In CV, we use the CIFAR-10 and CIFAR-100 datasets (Krizhevsky et al., 2009). Each dataset contains 45,000 training, 5000 validation, and 10,000 test images. In NLP, we use the IMDB dataset (Maas et al., 2011), with 20,000 training, 5,000 validation, and 25,000 test samples; and the CoLA dataset (Warstadt et al., 2019), with 6,355 training, 907 validation, and 1,816 test samples.

**Baselines**. We compare DIRECTOR with various uncertainty-aware baselines: Temperature Scaling (TS) (Guo et al., 2017), Monte Carlo Dropout (MCD) (Gal & Ghahramani, 2016), Stochastic Variational Deep Kernel Learning (SV-DKL) (Wilson et al., 2016a), Kronecker-Factored Last-Layer Laplace Approximation (KFLLA) (Kristiadi et al., 2020), Sparse Gaussian Process Attention (SGPA) (Chen & Li, 2023), and Kernel-Eigen Pair Sparse Variational Gaussian Processes Attention (KEP-SVGP or KEP for brevity) (Chen et al., 2024c).

**Pre-Trained Models.** We conduct our uncertainty-aware reconfiguration experiments on two pre-trained architectures which include (i) vanilla transformer-based model and (ii) uncertainty-aware transformer KEP (Chen et al., 2024b). For each architecture, we pre-train a 7-layer vision transformer (ViT (Dosovitskiy et al., 2020)) for experiments on CIFAR-10 and CIFAR-100, and a 5-layer transformer (Vaswani et al., 2017) for CoLA and IMDB. We also evaluate variants of uncertainty-aware transformer KEP (Chen et al., 2024b), where the first $n - k$ attention blocks use standard MHSA and the last $k$ blocks use GP-reparameterized attention; we denote this variant as KEP-$k/n$.

**Unified Transition Model.** We parameterize the unified spatiotemporal transition model in our diffusion-based reparameterization using a single-block DiT (Peebles & Xie, 2022), with embedding dimensions matched to those of the pre-trained models (384 for CIFAR, 256 for CoLA, and 128 for IMDB). It is configured with 12 attention heads for CIFAR-10/100, 8 for CoLA, and 4 for IMDB which are followed by a feed-forward network incorporating adaptive LN for timestep embedding. This is configured to contain fewer parameters than the original pre-trained backbone to improve memory efficiency. Our transition model comprises only 2.7M parameters compared to 6.24M in ViT for vision tasks, and 2.59M compared to 3.38M in the text transformer for NLP task.

**Evaluation Metrics.** For in-distribution classification, we evaluate predictive performance using accuracy (ACC) for CIFAR-10, CIFAR-100, and IMDB, and Matthew's Correlation Coefficient (MCC) for CoLA. Calibration is assessed with Negative Log-Likelihood (NLL $\times$ 10), Expected Calibration Error (ECE %), and Brier Score (%). Failure prediction is measured using Area Under the Risk-Coverage Curve (AURC %), Area Under the Receiver Operating Characteristic Curve (AUROC %), and False Positive Rate at 95% True Positive Rate (FPR95 %).

For out-of-distribution (OOD) robustness, the same metrics are applied to the CIFAR-10-C and CoLA-OOD datasets. OOD detection performance is quantified using AUROC % and Area Under the Precision-Recall Curve (AUPR %). All metrics are reported as mean $\pm$ standard error over five runs. All experiments are conducted on a single NVIDIA L40 GPU.

## 3.2 RESULTS AND DISCUSSION

### 3.2.1 IN-DISTRIBUTION CLASSIFICATION

**Comparison with Pre-Trained Models.** DIRECTOR demonstrates superior performance compared to pre-trained models across nearly all tasks and metrics (Table 1). While DIRECTOR does not outperform KEP-7/7 in ECE and NLL on CIFAR-10 or in ECE on CIFAR-100, the differences are marginal (ECE < 2%, NLL < 0.1). Conversely, DIRECTOR achieves significantly higher accuracy than KEP-7/7 (87.06% versus 82.68% on CIFAR-10; 60.85% versus 57.06% on CIFAR-100), a lower Brier score, and outperforms KEP-7/7 in failure prediction tasks. These results demonstrate that DIRECTOR not only provides better-calibrated uncertainty estimates but also enhances predictive accuracy, underscoring the effectiveness of our propagation-based model for UQ.

**Comparison with Existing Uncertainty-Aware Baselines.** Beyond vanilla pre-trained models, DIRECTOR also surpasses other uncertainty-aware approaches in both predictive performance and uncertainty quantification across multiple tasks (Table 2). For a fair comparison, DIRECTOR and KEP are configured with optimal settings and evaluated against standard baselines. Overall, DIREC-

Table 1: Performance comparison between pre-trained transformers and their diffusion-inspired reconfiguration using DIRECTOR on in-distribution classification tasks. KEP-$k/n$ denotes a pre-trained transformer using GP-reparameterized architecture (KEP (Chen et al., 2024c)) for the last $k$ attention blocks and standard MHSA for the remaining blocks. Better results are shown in **bold**.

| Dataset | Method | ACC/MCC ↑ | AURC ↓ | AUROC ↑ | FPR95 ↓ | ECE ↓ | NLL ↓ | Brier ↓ |
|---|---|---|---|---|---|---|---|---|
| CIFAR-10 | ViT | 83.84±0.09 | 4.05±0.11 | 86.42±0.37 | 67.13±1.98 | 12.51±0.20 | 10.91±0.39 | 28.03±0.15 |
| | DIRECTOR | **85.67**±0.88 | **3.10**±0.45 | **87.90**±1.04 | **61.92**±1.85 | **9.67**±0.62 | **6.94**±0.52 | **23.54**±1.46 |
| | KEP-1/7 | 84.52±0.25 | 3.52±0.11 | 87.52±0.27 | 65.14±1.27 | 10.93±0.26 | 8.21±0.15 | 25.80±0.40 |
| | DIRECTOR | **86.51**±0.57 | **2.78**±0.16 | **88.28**±0.16 | **61.85**±1.93 | **8.61**±0.49 | **6.11**±0.27 | **21.90**±0.82 |
| | KEP-7/7 | 82.68±0.12 | 4.46±0.05 | 85.71±0.34 | 66.90±1.78 | **6.95**±0.36 | **5.89**±0.11 | 25.90±0.23 |
| | DIRECTOR | **87.06**±0.18 | **2.57**±0.11 | **88.60**±0.49 | **61.74**±0.52 | 8.42±0.15 | 5.96±0.18 | **21.11**±0.32 |
| CIFAR-100 | ViT | 52.94±0.63 | 22.88±0.64 | 81.07±0.56 | 75.96±2.39 | 30.73±0.61 | 32.71±0.91 | 74.72±1.20 |
| | DIRECTOR | **57.79**±0.57 | **18.58**±0.37 | **82.60**±0.11 | **71.50**±1.42 | **22.31**±0.49 | **22.16**±0.36 | **62.57**±0.85 |
| | KEP-1/7 | 55.74±0.77 | 20.20±0.64 | 82.10±0.20 | 73.82±0.92 | 27.07±0.71 | 27.45±0.62 | 68.54±1.18 |
| | DIRECTOR | **58.78**±1.52 | **17.63**±1.08 | **82.99**±0.41 | **71.40**±1.24 | **21.17**±1.03 | **21.15**±0.93 | **60.96**±1.73 |
| | KEP-7/7 | 57.06±0.56 | 19.38±0.60 | 82.02±0.39 | 72.78±0.69 | **21.31**±3.85 | 22.41±2.29 | 63.07±2.11 |
| | DIRECTOR | **60.85**±2.98 | **16.35**±2.34 | **82.91**±0.67 | **71.46**±1.43 | 21.43±2.09 | **20.55**±2.42 | **58.91**±4.63 |
| IMDB | Transformer | 85.59±0.50 | 4.73±0.32 | 80.75±0.55 | 75.45±1.10 | 6.96±0.24 | 3.95±0.44 | 22.28±1.26 |
| | DIRECTOR | **86.07**±0.61 | **4.57**±0.39 | **80.84**±0.67 | **74.24**±0.87 | **5.40**±1.90 | **3.60**±0.34 | **21.08**±1.32 |
| | KEP-1/5 | 85.76±0.71 | 4.54±0.42 | 81.02±0.70 | 74.87±0.87 | 5.51±2.94 | 3.79±0.51 | 21.62±1.42 |
| | DIRECTOR | **87.13**±0.19 | **4.07**±0.30 | **81.55**±0.62 | **73.35**±0.14 | **3.16**±2.54 | **3.24**±0.31 | **19.31**±0.97 |
| | KEP-5/5 | 84.57±0.81 | 5.48±0.60 | 79.32±1.25 | 77.03±1.48 | 7.83±3.28 | 5.17±2.13 | 24.23±2.56 |
| | DIRECTOR | **85.74**±0.34 | **4.58**±0.23 | **80.95**±0.42 | **75.08**±1.15 | **2.33**±1.54 | **3.36**±0.12 | **20.70**±0.65 |
| CoLA | Transformer | 29.92±1.17 | 20.80±1.21 | 64.22±1.46 | 90.01±2.84 | 26.44±1.90 | 19.66±4.18 | 55.09±2.68 |
| | DIRECTOR | **31.85**±2.46 | **19.74**±1.62 | **64.52**±1.71 | **89.52**±3.61 | **23.94**±0.49 | **14.29**±3.01 | **50.82**±1.19 |
| | KEP-1/5 | 30.86±2.03 | 19.86±2.04 | 65.18±1.83 | **89.10**±2.60 | 24.98±2.08 | 16.28±4.35 | 52.86±2.88 |
| | DIRECTOR | 31.84±1.88 | **19.42**±1.83 | **65.54**±1.78 | 90.37±0.89 | **13.77**±6.58 | **8.26**±3.75 | **43.54**±3.65 |
| | KEP-5/5 | 29.28±1.21 | 20.82±1.98 | 64.47±0.90 | 89.65±1.04 | 18.95±5.66 | 11.83±7.96 | 48.65±5.84 |
| | DIRECTOR | **32.05**±1.56 | **18.69**±1.39 | **64.71**±1.37 | **89.53**±1.67 | **12.88**±5.74 | **7.68**±1.70 | **42.20**±2.50 |

TOR achieves the **highest** performance in 21/28 settings (across 4 datasets and 7 UQ/performance metrics) and the **second-highest** in 4/28 settings, establishing a new state-of-the-art in UQ.

### 3.2.2 DISTRIBUTION SHIFT ROBUSTNESS

We also assess both the uncertainty quantification and predictive performance of DIRECTOR in scenarios with distribution shifts in both image classification and linguistic acceptability tasks. For vision, we use the CIFAR-10-C dataset, which includes 15 corruption types (e.g., noise, blur) at 5 severity levels (Hendrycks & Dietterich, 2019). For language, we use the CoLA OOD dataset, which assesses novel linguistic structures (Warstadt et al., 2019). On CIFAR-10-C (see Table 3), DIRECTOR achieves better performance than KEP-7/7 in all metrics. It also remains competitive with ViT and KEP-1/7 in predictive performance while improving on calibration metrics. On CoLA OOD (see Table 4), DIRECTOR significantly improves calibration metrics without sacrificing MCC, except when compared to KEP-5/5 which has slightly better MCC but is weaker on UQ metrics. These observations consistently demonstrate both the performance robustness and generalization capability of DIRECTOR under test scenarios with distribution shifts.

### 3.2.3 OUT-OF-DISTRIBUTION DETECTION

Uncertainty-aware baselines can also be evaluated in terms of their abilities to distinguish between (i) correctly classified in-distribution samples, (ii) misclassified in-distribution samples, and (iii) out-of-distribution (OOD) samples. To assess this capability, we report the average performance with standard deviation of DIRECTOR in a number of OOD detection scenarios (see Table 5) using the AUROC/AUPR metrics and two standard methods: (1) Maximum Softmax Probability (Hendrycks & Gimpel, 2017) and Entropy Maximization (Chan et al., 2021). Using CIFAR-10 as the in-distribution dataset, our evaluation on SVHN, CIFAR-100, and LSUN demonstrates that the performance of most pre-trained transformer (using CIFAR-100 data) in most cases can be substantially improved by their corresponding diffusion-based reconfiguration. Notably, when the diffusion-based reconfiguration of KEP-2/7 produced by DIRECTOR outperforms all baselines as highlighted in blue.

Table 2: Comparison of average performance achieved by the diffusion-based reconfigured KEP (Chen et al., 2024c) produced by DIRECTOR and other uncertainty-aware baselines on in-distribution classification tasks. **Blue** marks the best result across all baselines for a dataset while **brown** denotes the second-best. ↑ indicates that higher values are better, while ↓ indicates that lower values are better.

| Dataset | Method | ACC/MCC ↑ | AURC ↓ | AUROC ↑ | FPR95 ↓ | ECE ↓ | NLL ↓ | Brier ↓ |
|---|---|---|---|---|---|---|---|---|
| CIFAR-10 | TS | 83.84±0.09 | 3.88±0.10 | 86.82±0.37 | 65.99±1.94 | 9.22±0.36 | 6.58±0.16 | 25.50±0.13 |
| | MCD | 84.06±0.23 | 8.65±0.03 | 86.51±0.32 | 66.15±0.60 | 9.47±0.16 | 8.36±0.32 | 25.45±0.29 |
| | KFLLA | 83.84±0.10 | 3.91±0.11 | 86.71±0.45 | 65.44±1.58 | 8.18±0.80 | 6.09±0.40 | 24.98±0.59 |
| | SV-DKL | 83.23±0.17 | 4.39±0.18 | 85.94±0.36 | 66.96±1.30 | 11.64±0.81 | 9.85±1.09 | 27.97±0.77 |
| | SGPA | 75.59±3.63 | 8.41±2.37 | 82.65±1.71 | 71.78±2.73 | 1.92±0.55 | 7.11±0.95 | 33.98±4.57 |
| | ViT | 83.84±0.09 | 4.05±0.11 | 86.42±0.37 | 67.13±1.98 | 12.51±0.20 | 10.91±0.39 | 28.03±0.15 |
| | KEP | 84.52±0.25 | 3.52±0.11 | 87.52±0.27 | 65.14±1.27 | 10.93±0.26 | 8.21±0.15 | 25.80±0.40 |
| | DIRECTOR | 87.06±0.18 | 2.57±0.11 | 88.60±0.49 | 61.74±0.52 | 8.42±0.15 | 5.96±0.18 | 21.11±0.32 |
| CIFAR-100 | TS | 52.94±0.63 | 22.34±0.61 | 82.29±0.48 | 71.65±1.98 | 17.06±0.42 | 21.57±0.52 | 64.77±0.95 |
| | MCD | 53.49±0.62 | 22.24±0.56 | 81.60±0.19 | 73.02±0.51 | 25.93±0.37 | 29.24±0.73 | 70.02±0.92 |
| | KFLLA | 52.27±0.86 | 23.96±0.78 | 81.30±0.48 | 71.42±1.92 | 18.52±5.40 | 20.89±0.57 | 66.51±1.66 |
| | SV-DKL | 51.03±0.60 | 24.38±0.43 | 81.32±0.55 | 73.99±1.40 | 25.46±0.72 | 28.93±0.66 | 71.90±0.74 |
| | SGPA | 52.77±0.52 | 22.84±0.52 | 81.65±0.36 | 72.02±1.74 | 10.33±2.25 | 19.10±0.57 | 62.08±1.04 |
| | ViT | 52.94±0.63 | 22.88±0.64 | 81.07±0.56 | 75.96±2.39 | 30.73±0.61 | 32.71±0.91 | 74.72±1.20 |
| | KEP | 57.06±0.56 | 19.38±0.60 | 82.02±0.39 | 72.78±0.69 | 21.31±3.85 | 22.41±2.29 | 63.07±2.11 |
| | DIRECTOR | 60.85±2.98 | 16.35±2.34 | 82.91±0.67 | 71.46±1.43 | 21.43±2.09 | 20.55±2.42 | 58.91±4.63 |
| IMDB | TS | 85.59±0.50 | 4.73±0.32 | 80.75±0.55 | 75.45±1.10 | 2.91±1.51 | 3.41±0.13 | 21.04±0.74 |
| | MCD | 85.96±0.42 | 4.40±0.24 | 81.40±0.55 | 74.79±0.88 | 4.18±2.03 | 3.47±0.23 | 20.72±0.82 |
| | KFLLA | 85.59±0.50 | 4.71±0.30 | 80.82±0.48 | 75.45±1.11 | 5.84±2.21 | 6.93±0.00 | 21.86±1.19 |
| | SV-DKL | 85.69±0.66 | 5.58±0.79 | 78.54±2.20 | 75.32±0.84 | 8.52±1.57 | 4.49±0.65 | 23.10±1.66 |
| | SGPA | 85.39±0.36 | 4.96±0.49 | 80.04±1.14 | 76.44±0.96 | 6.04±1.71 | 3.96±0.50 | 22.19±1.06 |
| | Transformer | 85.59±0.50 | 4.73±0.32 | 80.75±0.55 | 75.45±1.10 | 6.96±2.05 | 3.95±0.44 | 22.28±1.26 |
| | KEP | 85.76±0.71 | 4.54±0.42 | 81.02±0.70 | 74.87±0.87 | 5.51±2.94 | 3.79±0.51 | 21.62±1.42 |
| | DIRECTOR | 87.13±0.19 | 4.07±0.30 | 81.55±0.62 | 73.35±0.14 | 3.16±2.54 | 3.24±0.31 | 19.31±0.97 |
| CoLA | TS | 29.92±1.17 | 20.84±1.23 | 64.31±1.44 | 89.93±2.95 | 23.22±2.99 | 11.04±1.91 | 51.70±3.42 |
| | MCD | 30.04±1.02 | 20.66±1.11 | 64.53±1.00 | 89.51±1.35 | 24.96±1.79 | 17.83±3.61 | 53.52±2.59 |
| | KFLLA | 29.89±1.14 | 20.82±1.26 | 64.22±1.46 | 89.87±3.24 | 24.36±2.25 | 12.16±1.49 | 52.80±2.85 |
| | SV-DKL | 30.07±1.41 | 22.76±2.28 | 61.98±3.09 | 89.00±2.55 | 25.71±1.60 | 17.96±3.26 | 54.40±2.13 |
| | SGPA | 31.53±2.05 | 20.44±2.60 | 64.34±1.95 | 90.79±0.87 | 26.22±1.51 | 28.65±7.23 | 54.08±2.44 |
| | Transformer | 29.92±1.17 | 20.80±1.21 | 64.22±1.46 | 90.01±2.84 | 26.44±1.90 | 19.66±4.18 | 55.09±2.68 |
| | KEP | 30.86±2.03 | 19.86±2.04 | 65.18±1.83 | 89.10±2.60 | 24.98±2.08 | 16.28±4.35 | 52.86±2.88 |
| | DIRECTOR | 32.05±1.56 | 18.69±1.39 | 64.71±1.37 | 89.53±1.67 | 12.88±5.74 | 7.68±1.70 | 42.20±2.50 |

Table 3: Comparison on CIFAR10-C.

| Method | ACC ↑ | ECE ↓ | NLL ↓ | Brier ↓ |
|---|---|---|---|---|
| ViT | 69.67±0.34 | 24.30±0.31 | 23.59±1.00 | 53.07±0.59 |
| DIRECTOR | 68.89±1.44 | 22.32±1.09 | 17.71±1.02 | 51.77±2.39 |
| KEP-1/7 | 69.87±0.45 | 22.12±0.47 | 18.54±0.63 | 50.65±0.90 |
| DIRECTOR | 69.29±0.66 | 20.98±0.65 | 16.23±0.60 | 50.07±1.20 |
| KEP-7/7 | 59.57±0.30 | 21.78±0.59 | 17.17±0.39 | 60.67±0.72 |
| DIRECTOR | 68.12±0.26 | 22.19±0.19 | 17.70±0.17 | 52.27±0.35 |

Table 4: Comparison on CoLA OOD.

| Method | MCC ↑ | ECE ↓ | NLL ↓ | Brier ↓ |
|---|---|---|---|---|
| Transformer | 18.43±3.55 | 31.99±2.70 | 23.82±5.16 | 65.50±4.32 |
| DIRECTOR | 23.06±4.69 | 28.72±1.87 | 17.18±3.75 | 59.91±2.20 |
| KEP-1/5 | 19.44±1.94 | 30.33±1.48 | 19.67±4.82 | 61.97±2.25 |
| DIRECTOR | 22.10±5.49 | 17.50±7.52 | 9.34±4.50 | 49.92±5.80 |
| KEP-5/5 | 21.14±3.48 | 23.27±6.61 | 14.25±10.66 | 55.12±7.78 |
| DIRECTOR | 20.20±5.46 | 17.49±5.93 | 8.61±1.99 | 48.55±3.67 |

Additional results, including deep ensembles (Lakshminarayanan et al., 2017), a large-scale ViT-B-16 (86M) experiment, and loss component ablations, are provided in Appendix A.6.

# 4 RELATED WORK

In safety-critical decision-making applications (e.g., healthcare (A. et al., 2021; Lopez et al., 2023; Band et al., 2022)), models must recognize when their confidence is low to defer decisions to human experts (Pietraszek, 2007; Tran et al., 2022b). However, existing transformers typically ignore uncertainty due to point-estimate designs throughout their stack of neural transformations (Papamarkou et al., 2024). Prior investigation in Bayesian deep learning (BDL) are often restricted to moderate-sized DL architectures (Wang & Yeung, 2016; Mukhoti & Gal, 2018; Kendall & Gal, 2017; Gustafsson et al., 2020; Chien & Ku, 2015; Ritter et al., 2021; Tran et al., 2019; Fortuin

Table 5: Comparison of average OOD detection performance in AUROC (%) and AUPR (%) (with reported standard deviation) achieved by the tested baselines over 5 independent runs.

| Method | SVHN | | CIFAR-100 | | LSUN | |
|---|---|---|---|---|---|---|
| | AUROC ↑ | AUPR ↑ | AUROC ↑ | AUPR ↑ | AUROC ↑ | AUPR ↑ |
| MCD | 87.09±8.53 | 91.46±4.87 | 76.27±0.35 | 78.82±0.41 | 88.41±2.05 | 91.19±1.51 |
| KFLLA | 89.47±9.07 | 92.92±5.27 | 77.27±0.42 | 79.88±0.42 | 90.77±2.93 | 92.61±2.24 |
| SVDKL | 86.59±6.86 | 90.78±4.04 | 75.99±0.74 | 77.89±1.23 | 87.81±2.52 | 90.60±1.91 |
| SGPA | 61.57±5.11 | 74.59±3.50 | 73.42±1.87 | 75.93±1.87 | 67.34±9.77 | 76.76±5.93 |
| ViT | **87.09**±8.53 | **91.46**±4.87 | 76.27±0.35 | 78.82±0.41 | **88.41**±2.05 | **91.19**±1.51 |
| DIRECTOR | 83.19±10.94 | 88.84±6.37 | **78.57**±0.94 | **81.33**±0.95 | 83.97±7.19 | 88.70±4.50 |
| KEP-1/7 | 75.28±19.12 | 81.92±13.36 | 77.93±0.39 | 80.85±0.46 | 85.64±4.47 | 88.98±3.57 |
| DIRECTOR | **90.73**±4.07 | **93.21**±2.78 | **79.29**±0.50 | **82.11**±0.40 | **89.38**±3.22 | **92.08**±2.37 |
| KEP-2/7 | 88.25±4.67 | 91.56±3.14 | 77.71±0.55 | 80.58±0.53 | 88.35±3.62 | 91.18±2.75 |
| DIRECTOR | 92.14±5.70 | 94.49±3.59 | 79.43±0.57 | 82.15±0.57 | 91.39±2.74 | 93.63±1.86 |
| KEP-7/7 | 77.16±1.62 | 84.09±1.28 | 76.21±0.41 | 78.82±0.41 | 77.01±3.28 | 82.42±2.73 |
| DIRECTOR | **79.33**±20.83 | **85.92**±13.88 | **79.11**±0.39 | **81.70**±0.38 | **86.84**±4.07 | **90.05**±3.16 |

et al., 2022; Tran et al., 2020; Rudner et al., 2022a; Qiu et al., 2023), limiting scalability to large networks (Papamarkou et al., 2024). To elaborate, we next discuss two main UQ paradigms:

**UQ integration during training.** These methods treat model parameters as random variables and learn their posterior distributions conditioned on data. Bayesian neural networks approximate posteriors via MCMC or variational inference (Blundell et al., 2015; Guo et al., 2022), while deep ensembles (Lakshminarayanan et al., 2017) approximate them non-parametrically through diverse initializations. Evidential methods (Wilson et al., 2016b; Sensoy et al., 2018) map features to prior parameters over the likelihood for a closed-form uncertainty estimation via conjugate priors. Sampling-based methods offer higher fidelity by avoiding structural assumptions but incur prohibitive sampling cost for large models (Wenzel et al., 2020). In contrast, variational and evidential approaches are more scalable but less accurate due to biased approximations and restrictive parameterizations (Wilson et al., 2022; Chen et al., 2015). Overall, these approaches approximate or sample from the parameter posterior, which remains highly intractable and often yields unreliable estimates.

**UQ integration post-training.** These methods recalibrate prediction confidence by augmenting a trained model's output without altering most its parameters, including data augmentation (Wang et al., 2019), Monte Carlo dropout (Gal & Ghahramani, 2016), and input-gradient norms (Ash et al., 2019). There are also learning-based approaches that adjust output probabilities to better reflect correctness, such as temperature scaling (Guo et al., 2017), replacing the solution head with probabilistic alternatives (e.g., Gaussian processes (Rasmussen & Williams, 2006), SNGPs (Liu et al., 2020; Bradshaw et al., 2017)), or Laplace approximation that fits a local Gaussian approximation to the weight posterior around the model's learned parameters (Li et al., 2023). More recently, conformal prediction (Marx et al., 2022) offers a black-box calibration method that uses a pre-trained model's softmax scores and test-time data to produce prediction sets with marginal coverage guarantees.

## 5 CONCLUSION

We introduce a diffusion-inspired reconfiguration of pre-trained transformers that enables principled uncertainty propagation across the entire feature transformation stack. Our approach builds on the established connection between multi-head self-attention (MHSA) and Gaussian process (GP) prediction, reparameterizing the feature transformation stack as a sequence of neuralized Gaussian transitions. This sequence is then distilled into a diffusion process with a learnable unified spatiotemporal transition model mapping between the data and feature distributions, thereby embedding expressive uncertainty-aware structure within the original transformer while preserving its predictive performance. We comprehensively evaluate this approach across diverse vision and language tasks, consistently demonstrating its effectiveness. These results point toward a new direction for embed-

ding probabilistic reasoning into the internal structure of large pre-trained models, enhancing their reliability in risk-sensitive applications and revealing a new paradigm shift to scalable UQ.

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

# A    APPENDIX / SUPPLEMENTAL MATERIAL

## A.1    MULTI-HEAD SELF-ATTENTION

**Self-Attention.** Given an input of the attention layer $\mathbf{U} \in \mathbb{R}^{N \times d}$, where $N$ is the number of data points and $d$ is the embedding dimension, self-attention computes queries, keys, and values via $\mathbf{Q} = \mathbf{U}\mathbf{W}_q$, $\mathbf{K} = \mathbf{U}\mathbf{W}_k$, and $\mathbf{V} = \mathbf{U}\mathbf{W}_v$, with projection matrices $\mathbf{W}_q, \mathbf{W}_k, \mathbf{W}_v \in \mathbb{R}^{d \times d_h}$, where $d_h$ is projected dimension. The output of the self-attention is:

$$\mathbf{F} \quad = \quad \text{softmax}\left(\frac{\mathbf{Q}\mathbf{K}^\top}{\sqrt{d_h}}\right)\mathbf{V} = \mathbf{A}_{qk}\mathbf{V}, \tag{13}$$

where attention matrix $\mathbf{A}_{qk} \in \mathbb{R}^{N \times N}$ encodes the pairwise similarity between the queries and keys.

**Multi-Head Self-Attention (MHSA).** MHSA employs $n$ parallel attention heads, each independently computing a self-attention output $\mathbf{F}^{(h)}$ with corresponding $\mathbf{W}_q^{(h)}, \mathbf{W}_k^{(h)}, \mathbf{W}_v^{(h)} \in \mathbb{R}^{d \times d_h}$ as defined in Eq. 13. To maintain computational efficiency, the dimensionality of each head is typically set to $d_h = d/n$. The outputs from all attention heads are concatenated and subsequently projected back to the input dimension, forming MHSA's output:

$$\mathbf{R} = \mathbf{O}\left[\mathbf{F}^{(1)}, \mathbf{F}^{(2)}, \dots, \mathbf{F}^{(n)}\right] \tag{14}$$

where $\mathbf{O} \in \mathbb{R}^{d \times (n \cdot d_h)}$ is a projection matrix.

## A.2    SELF-ATTENTION AS GAUSSIAN PROCESS INFERENCE

**Kernel Attention or K-Attention (Tsai et al., 2019)** has extended the attention mechanism by replacing cosine similarity with a general kernel function $\kappa(\cdot, \cdot) : \mathbb{R}^d \times \mathbb{R}^d \to \mathbb{R}$ to compute pairwise similarities. Specifically, the attention matrix $\mathbf{A}_{qk}$ is replaced by a kernel matrix $\mathcal{K}_{qk}$, where each entry is defined as $[\mathcal{K}_{qk}]_{i,j} = \kappa(\mathbf{U}_{i,:}, \mathbf{U}_{j,:})$. Consequently, the output of the kernel attention mechanism is:

$$\mathbf{F} = \mathcal{K}_{qk}\mathbf{V}, \tag{15}$$

**Sparse Gaussian Process Attention (SGPA) (Chen & Li, 2023)** leverages the Sparse Variational Gaussian Process (SVGP) framework to approximate posterior variances in the attention mechanism. For each dimension $i$ of attention output, the posterior mean and covariance are given by:

$$\mu_i = \mathcal{K}_{qk}\mathbf{V}_{:,i}\,, \text{ and } \Sigma_i = \mathcal{K}_{qq} + \mathcal{K}_{qk}\left(\mathcal{K}_{kk}^{-1}[\mathcal{S}]_{:,:,i}\mathcal{K}_{kk}^{-1} - \mathcal{K}_{kk}^{-1}\right)\mathcal{K}_{kq} \tag{16}$$

where $\mathcal{S} \in \mathbb{R}^{N \times N \times d_h}$ is a set of variational covariance parameters, optimized via the SVGP evidence lower bound. The output for each dimension $i$ is then sampled using the reparameterization trick:

$$\mathbf{F}_{:,i} = \mu_i + \Sigma_i^{1/2} \cdot \epsilon_i, \quad \epsilon_i \sim \mathcal{N}(0, I) \tag{17}$$

By stacking the results across all $d_h$ output dimensions, the final output of the attention head is:

$$\mathbf{F} = \left[\mu_1 + \Sigma_1^{1/2}\epsilon_1\,, \ \cdots\,, \ \mu_{d_h} + \Sigma_{d_h}^{1/2}\epsilon_{d_h}\right] \in \mathbb{R}^{N \times d_h} \tag{18}$$

**Kernel-Eigen Pair Sparse Variational Gaussian Processes Attention (KEP-SVGP) (Chen et al., 2024c).** The constraint of imposing a symmetric kernel matrix $\mathcal{K}_{qk}$ in **K-Attention** and **SGPA** (requiring $\mathbf{W}_q = \mathbf{W}_k$) can restrict the model's representation capacity by eliminating the inherent asymmetry of the original attention matrix $\mathbf{A}_{qk}$. To overcome this restriction, **KEP-SVGP** employs two Gaussian Processes (GPs), leveraging the symmetry of $\mathcal{K}_{qk}\mathcal{K}_{qk}^\top$ and $\mathcal{K}_{qk}^\top\mathcal{K}_{qk}$. Each attention output dimension is then computed by combining the contributions from the two GPs.

More specifically, building on the KSVD framework and the Primal-Attention formulation (Chen et al., 2024a), KEP-SVGP introduces two sets of $s$-dimensional attention outputs to model the left and right eigenspaces, denoted as $F_{[i]}^e := F^e[:, i]$ and $F_{[i]}^r := F^r[:, i] \in \mathbb{R}^N$ for $i = 1, \dots, s$, corresponding to the primal features $e(\mathbf{U})$ and $r(\mathbf{U})$ in (Chen et al., 2024a), respectively. To model

these outputs, the following SVGP priors are defined based on the induced symmetric kernels $\mathcal{K}_{qk}\mathcal{K}_{qk}^\top$ and $\mathcal{K}_{qk}^\top\mathcal{K}_{qk}$:

$$\begin{pmatrix} \mathbf{F}_i^e \\ \mathbf{u}_i^e \end{pmatrix} \sim \mathcal{GP}\left( \mathbf{0}, \begin{pmatrix} \mathcal{K}_{qk}\mathcal{K}_{qk}^\top & \mathbf{H}_e\Lambda^2 \\ \Lambda^2\mathbf{H}_e^\top & \Lambda^2 \end{pmatrix} \right), \quad \begin{pmatrix} \mathbf{F}_i^r \\ \mathbf{u}_i^r \end{pmatrix} \sim \mathcal{GP}\left( \mathbf{0}, \begin{pmatrix} \mathcal{K}_{qk}^\top\mathcal{K}_{qk} & \mathbf{H}_r\Lambda^2 \\ \Lambda^2\mathbf{H}_r^\top & \Lambda^2 \end{pmatrix} \right), \quad (19)$$

where $\Lambda \in \mathbb{R}^{s\times s}$ is a diagonal matrix of the top-$s$ singular values of $\mathcal{K}_{qk}$, and $\mathbf{H}_e, \mathbf{H}_r \in \mathbb{R}^{N\times s}$ contain the corresponding top-$s$ left and right singular vectors, respectively. Using variational distributions $\mathbf{u}_{[i]}^e, \mathbf{u}_{[i]}^r \sim \mathcal{N}(\mathbf{m}_{\mathbf{u},[i]}, S_{\mathbf{uu},[i]})$, closed-form posteriors $q(\mathbf{F}_i^c|\mathbf{U}) = \int q(\mathbf{F}_i^c|\mathbf{u}_i)q(\mathbf{u}_i)\,\mathrm{d}\mathbf{u}_i$ ($c \in \{e, r\}$) are derived as:

$$q(\mathbf{F}_i^e|\mathbf{U}) \sim \mathcal{N}\Big( \underbrace{E_U\Lambda^{-1}\mathbf{m}_{\mathbf{u},[d]}}_{\mu^e:=\mathbf{m}_{[i]}^e}, \underbrace{E_U\Lambda^{-2}S_{\mathbf{uu},[i]}E_U^\top}_{\Sigma^e:=\mathbf{L}_{[i]}^e{\mathbf{L}_{[i]}^e}^\top} \Big),$$

$$q(\mathbf{F}_i^r|\mathbf{U}) \sim \mathcal{N}\Big( \underbrace{R_U\Lambda^{-1}\mathbf{m}_{\mathbf{u},[d]}}_{\mu^r:=\mathbf{m}_{[i]}^r}, \underbrace{R_U\Lambda^{-2}S_{\mathbf{uu},[i]}R_U^\top}_{\Sigma^r:=\mathbf{L}_{[i]}^r{\mathbf{L}_{[i]}^r}^\top} \Big), \quad (20)$$

where $E_U := [e(\mathbf{U}_i), \ldots, e(\mathbf{U}_N)]^\top \in \mathbb{R}^{N\times s}$ and $R_U := [r(\mathbf{U}_i), \ldots, r(\mathbf{U}_N)]^\top \in \mathbb{R}^{N\times s}$ are the projection matrices w.r.t. right and left singular vectors of KSVD in (Chen et al., 2024a). The variational parameters are $\mathbf{m}_{\mathbf{u}} \in \mathbb{R}^{s\times s}$, $S_{\mathbf{uu}} \in \mathbb{R}^{s\times s\times s}$ with the components $i$-th defined as $\mathbf{m}_{\mathbf{u},[i]} := \mathbf{m}_{\mathbf{u}}[:, i] \in \mathbb{R}^s$, and $S_{\mathbf{uu},[i]} := S_{\mathbf{uu}}[:, :, i] \in \mathbb{R}^{s\times s}$.

The outputs of the two SVGPs are sampled via the reparameterization trick:

$$F_{[i]}^e = \mathbf{m}_{[i]}^e + \mathbf{L}_{[i]}^e\epsilon, \quad F_{[i]}^r = \mathbf{m}_{[i]}^r + \mathbf{L}_{[i]}^r\epsilon, \text{ with } \epsilon \sim \mathcal{N}(0, I_N) \quad (21)$$

To fuse the outputs, two schemes are proposed: Addition ($F_{[i]}^{add} := F_{[i]}^e + F_{[i]}^r \in \mathbb{R}^N$) and Concatenation ($F_{[i]}^{cat} := [F_{[i]}^e; F_{[i]}^r] \in \mathbb{R}^{2N}$). To align with standard Transformer architecture, the $s$-dimensional attention outputs are linearly projected to the target dimension $d_h$. The final output $\mathbf{F} \in \mathbb{R}^{N\times d_h}$ is computed as: $\mathbf{F} = \mathbf{F}^{add}\mathbf{W}^{add}$ for the addition, $\mathbf{F} = \mathbf{W}_1^{cat}\mathbf{F}^{cat}\mathbf{W}_2^{cat}$ for the concatenation, where the projection matrices are $\mathbf{W}^{add} \in \mathbb{R}^{s\times d_h}$, $\mathbf{W}_1^{cat} \in \mathbb{R}^{N\times 2N}$ and $\mathbf{W}_2^{cat} \in \mathbb{R}^{s\times d_h}$.

### A.3 Denoising Diffusion Probabilistic Models

Denoising Diffusion Probabilistic Models (DDPMs) (Ho et al., 2020) transform data into noise through a gradual forward diffusion process and then learn to reverse this transformation. The forward process incrementally adds Gaussian noise to the data $\mathbf{X}_0$ over $T$ steps:

$$p(\mathbf{X}_t|\mathbf{X}_{t-1}) = \mathcal{N}(\mathbf{X}_t; \sqrt{1-\beta_t}\mathbf{X}_{t-1}, \beta_t\mathbf{I}), \quad (22)$$

where $\{\beta_t\}_{t=1}^T$ controls the noise schedule. This defines a Markov chain that progressively corrupts the data. The true reverse process $p(\mathbf{X}_{t-1}|\mathbf{X}_t)$ is generally intractable but becomes tractable when conditioned on $\mathbf{X}_0$:

$$p(\mathbf{X}_{t-1}|\mathbf{X}_t, \mathbf{X}_0) = \mathcal{N}\big(\mathbf{X}_{t-1}; \tilde{\mu}_t(\mathbf{X}_t, \mathbf{X}_0), \tilde{\beta}_t\mathbf{I}\big), \quad (23)$$

where closed-form expressions for $\tilde{\mu}_t$ and $\tilde{\beta}_t$ follow from Bayes' rule under the forward process. However, $\mathbf{X}_0$ is unknown at test time, the model instead learns a parameterized reverse process that conditions only on $(\mathbf{X}_t, t)$:

$$q_\theta(\mathbf{X}_{t-1}|\mathbf{X}_t) = \mathcal{N}(\mathbf{X}_{t-1}; \mu_\theta(\mathbf{X}_t, t), \Sigma_\theta(\mathbf{X}_t, t)), \quad (24)$$

Learning proceeds by minimizing the variational bound on the negative log-likelihood of the data, which encourages $q_\theta(\mathbf{X}_{t-1}|\mathbf{X}_t)$ to match the true reverse process $p(\mathbf{X}_{t-1}|\mathbf{X}_t, \mathbf{X}_0)$. This is typically implemented via noise prediction (score matching), where the network predicts the injected noise $\epsilon$ instead of $\mu_\theta$.

### A.4 Derivation of the upper bound of $L(\theta)$

The negative log-likelihood $L(\theta) = \mathbb{E}_{p(\mathbf{X}_0|\mathbf{X}_T)}\left[-\log q_\theta(\mathbf{X}_0 \mid \mathbf{X}_T)\right]$ is upper bounded by

$$L(\theta) \leq \mathbb{H}\Big(p(\mathbf{X}_0 \mid \mathbf{X}_T)\Big) + \sum_{t=1}^T \mathbb{E}_{p(\mathbf{X}_t|\mathbf{X}_T)}\Big[ D_{\mathrm{KL}}\Big(p(\mathbf{X}_{t-1}|\mathbf{X}_t) \,\|\, q_\theta(\mathbf{X}_{t-1} \mid \mathbf{X}_t)\Big)\Big] \quad (25)$$

*Proof.* From the definition of $L(\theta)$, we have

$$L(\theta) = -\mathbb{E}_{p(\mathbf{X}_0|\mathbf{X}_T)}\left[\log q_\theta(\mathbf{X}_0 \mid \mathbf{X}_T)\right] \tag{26}$$

$$= -\mathbb{E}_{p(\mathbf{X}_0|\mathbf{X}_T)}\left[\log\left(\int q_\theta(\mathbf{X}_{0:T-1} \mid \mathbf{X}_T)d\mathbf{X}_{1:T-1}\right)\right] \tag{27}$$

$$= -\mathbb{E}_{p(\mathbf{X}_0|\mathbf{X}_T)}\left[\log\left(\int p(\mathbf{X}_{1:T-1} \mid \mathbf{X}_0,\mathbf{X}_T)\frac{q_\theta(\mathbf{X}_{0:T-1} \mid \mathbf{X}_T)}{p(\mathbf{X}_{1:T-1} \mid \mathbf{X}_0,\mathbf{X}_T)}d\mathbf{X}_{1:T-1}\right)\right] \tag{28}$$

$$= -\mathbb{E}_{p(\mathbf{X}_0|\mathbf{X}_T)}\left[\log\left(\mathbb{E}_{p(\mathbf{X}_{1:T-1}|\mathbf{X}_0,\mathbf{X}_T)}\left[\frac{q_\theta(\mathbf{X}_{0:T-1} \mid \mathbf{X}_T)}{p(\mathbf{X}_{1:T-1} \mid \mathbf{X}_0,\mathbf{X}_T)}d\mathbf{X}_{1:T-1}\right]\right)\right] \tag{29}$$

$$\leq -\mathbb{E}_{p(\mathbf{X}_{0:T-1}|\mathbf{X}_T)}\log\left(\frac{q_\theta(\mathbf{X}_{0:T-1} \mid \mathbf{X}_T)}{p(\mathbf{X}_{1:T-1} \mid \mathbf{X}_0,\mathbf{X}_T)}\right) \quad \text{(Jensen's inequality)} \tag{30}$$

$$= \mathbb{E}_{p(\mathbf{X}_{0:T-1}|\mathbf{X}_T)}\log\left(\frac{p(\mathbf{X}_{1:T-1} \mid \mathbf{X}_0,\mathbf{X}_T)}{q_\theta(\mathbf{X}_{0:T-1} \mid \mathbf{X}_T)}\right) = L_{VLB} \tag{31}$$

Now, we derive the $L_{VLB}$ as follows:

$$L_{VLB} = \mathbb{E}_{p(\mathbf{X}_{0:T-1}|\mathbf{X}_T)}\left[\log\left(\frac{p(\mathbf{X}_{1:T-1} \mid \mathbf{X}_0,\mathbf{X}_T)}{q_\theta(\mathbf{X}_{0:T-1} \mid \mathbf{X}_T)}\right)\right] \tag{32}$$

$$= \mathbb{E}_{p(\mathbf{X}_{0:T-1}|\mathbf{X}_T)}\left[\log\left(\frac{p(\mathbf{X}_{0:T-1} \mid \mathbf{X}_T)}{q_\theta(\mathbf{X}_{0:T-1} \mid \mathbf{X}_T)p(\mathbf{X}_0 \mid \mathbf{X}_T)}\right)\right] \tag{33}$$

$$= \mathbb{E}_{p(\mathbf{X}_{0:T-1}|\mathbf{X}_T)}\left[-\log p(\mathbf{X}_0 \mid \mathbf{X}_T) + \log\left(\frac{p(\mathbf{X}_{0:T-1} \mid \mathbf{X}_T)}{q_\theta(\mathbf{X}_{0:T-1} \mid \mathbf{X}_T)}\right)\right] \tag{34}$$

$$= \mathbb{E}_{p(\mathbf{X}_{0:T-1}|\mathbf{X}_T)}\left[-\log p(\mathbf{X}_0 \mid \mathbf{X}_T) + \log\left(\prod_{t=1}^{T}\frac{p(\mathbf{X}_{t-1} \mid \mathbf{X}_t)}{q_\theta(\mathbf{X}_{t-1} \mid \mathbf{X}_t)}\right)\right] \tag{35}$$

$$= \mathbb{E}_{p(\mathbf{X}_{0:T-1}|\mathbf{X}_T)}\left[-\log p(\mathbf{X}_0 \mid \mathbf{X}_T) + \sum_{t=1}^{T}\log\left(\frac{p(\mathbf{X}_{t-1} \mid \mathbf{X}_t)}{q_\theta(\mathbf{X}_{t-1} \mid \mathbf{X}_t)}\right)\right] \tag{36}$$

$$= \mathbb{H}\left(p(\mathbf{X}_0 \mid \mathbf{X}_T)\right) + \sum_{t=1}^{T}\mathbb{E}_{p(\mathbf{X}_{0:T-1}|\mathbf{X}_T)}\left[\log\left(\frac{p(\mathbf{X}_{t-1} \mid \mathbf{X}_t)}{q_\theta(\mathbf{X}_{t-1} \mid \mathbf{X}_t)}\right)\right] \tag{37}$$

Additionally, we have:

$$\mathbb{E}_{p(\mathbf{X}_{0:T-1}|\mathbf{X}_T)}\left[\log\left(\frac{p(\mathbf{X}_{t-1} \mid \mathbf{X}_t)}{q_\theta(\mathbf{X}_{t-1} \mid \mathbf{X}_t)}\right)\right] \tag{38}$$

$$= \int p(\mathbf{X}_{0:T-1} \mid \mathbf{X}_T)\log\left(\frac{p(\mathbf{X}_{t-1} \mid \mathbf{X}_t)}{q_\theta(\mathbf{X}_{t-1} \mid \mathbf{X}_t)}\right)d\mathbf{X}_{0:T-1} \tag{39}$$

$$= \int p(\mathbf{X}_{t-1},\mathbf{X}_t \mid \mathbf{X}_T)\log\left(\frac{p(\mathbf{X}_{t-1} \mid \mathbf{X}_t)}{q_\theta(\mathbf{X}_{t-1} \mid \mathbf{X}_t)}\right)d\mathbf{X}_{t-1}d\mathbf{X}_t \tag{40}$$

$$= \int p(\mathbf{X}_t \mid \mathbf{X}_T)p(\mathbf{X}_{t-1} \mid \mathbf{X}_t)\log\left(\frac{p(\mathbf{X}_{t-1} \mid \mathbf{X}_t)}{q_\theta(\mathbf{X}_{t-1} \mid \mathbf{X}_t)}\right)d\mathbf{X}_{t-1}d\mathbf{X}_t \tag{41}$$

$$= \mathbb{E}_{p(\mathbf{X}_t|\mathbf{X}_T)}\left[D_{KL}\left(p(\mathbf{X}_{t-1} \mid \mathbf{X}_t) \,\|\, q_\theta(\mathbf{X}_{t-1} \mid \mathbf{X}_t)\right)\right] \tag{42}$$

A.5 PRACTICAL IMPLEMENTATION OF THE PROPOSED ALGORITHM IN EQ.12

The uncertainty-aware transition parameterization can be obtained via the objective in Eq.12:

$$\theta = \arg\min_{\theta}\left\{L_1(\theta) + L_2(\theta)\right\} \tag{43}$$

which combines the matching loss

$$L_1(\theta) = \mathbb{E}_{(\boldsymbol{X}_t,t)\sim p(\mathbf{X}_t|\mathbf{X}_T)}\left[D_{KL}\left(p(\boldsymbol{X}_{t-1} \mid \boldsymbol{X}_t) \,\|\, q_\theta(\boldsymbol{X}_{t-1} \mid \boldsymbol{X}_t)\right)\right], \tag{44}$$

with the performance-aware loss

$$L_2(\theta) \quad = \quad \mathop{\mathbb{E}}_{(\boldsymbol{X},\boldsymbol{y})\sim \boldsymbol{D}} \mathop{\mathbb{E}}_{\boldsymbol{X}_0 \sim q_\theta(\boldsymbol{X}_0 | \boldsymbol{X}_T)} \Big[ \mathrm{loss}\Big(\boldsymbol{X}_0, \boldsymbol{y}\Big)\Big], \tag{45}$$

where $\boldsymbol{X}$ is sampled from the training dataset $\boldsymbol{D}$ and is embedded with $\boldsymbol{X}_T = \mathrm{embed}(\boldsymbol{X})$. $\boldsymbol{X}_0$ is then sampled via iteratively simulating the current estimate of the probability path $q_\theta(\boldsymbol{X}_{t-1} \mid \boldsymbol{X}_t)$. In addition, the KL divergence in Eq. 44 can be reduced to the following loss,

$$D_{\mathrm{KL}}\Big(p \parallel q_\theta\Big) \quad \propto \quad \frac{1}{2}\left[\mathrm{tr}\left[\sigma_\theta^{-1}(\boldsymbol{X}_t, t)\sigma_t(\boldsymbol{X}_t)\right] + \log \frac{|\sigma_\theta(\boldsymbol{X}_t, t)|}{|\sigma_t(\boldsymbol{X}_t)|}\right] \tag{46}$$

$$+ \quad \frac{1}{2}\Big(m_\theta(\boldsymbol{X}_t, t) - m_t(\boldsymbol{X}_t)\Big)^\top \sigma_\theta^{-1}(\boldsymbol{X}_t)\Big(m_\theta(\boldsymbol{X}_t, t) - m_t(\boldsymbol{X}_t)\Big). \tag{47}$$

However, the KL computation still presents a challenge due to the high dimensionality of the covariance matrix $\sigma_t(\boldsymbol{X}_t)$, which complicates the evaluation of the trace and log-determinant terms. To mitigate this, we follow the approach of KEP by approximating $\sigma_t(\boldsymbol{X}_t)$ using a Cholesky-like factor $\mathbf{L}_t$ such that $\sigma_t(\boldsymbol{X}_t) = \mathbf{L}_t \mathbf{L}_t^\top$. For the parameterized covariance $\sigma_\theta(\boldsymbol{X}_t, t)$, we adopt a diagonal structure, making its Cholesky-like factor simply the element-wise square root of its diagonal. Matching these Cholesky-like factors ensures that the corresponding covariance matrices are aligned, effectively nullifying the trace and log-determinant terms in the KL divergence and enabling efficient optimization. Employing Cholesky-like factor and incorporating weighting terms yields the final objective:

$$\theta \quad = \quad \arg\min_\theta \ \Big\{\lambda_{\mathrm{mean}} L_{\mathrm{mean}}(\theta) + \lambda_{\mathrm{Cholesky}} L_{\mathrm{Cholesky}}(\theta) + \lambda_{\mathrm{NLL}} L_2(\theta)\Big\}, \tag{48}$$

where $\lambda_{\mathrm{mean}}$, $\lambda_{\mathrm{Cholesky}}$, and $\lambda_{\mathrm{NLL}}$ are weighting coefficients for the mean matching term, the Cholesky-like factor alignment, and the performance-aware loss, respectively. The individual loss components are defined as follows:

$$L_{\mathrm{mean}}(\theta) = \frac{1}{T}\sum_{t=1}^{T} \mathbb{E}_{p(\mathbf{X}_t | \mathbf{X}_T)} \left[\|m_\theta(\mathbf{X}_t, t) - m_t(\mathbf{X}_t)\|_2^2\right], \tag{49}$$

$$L_{\mathrm{Cholesky}}(\theta) = \frac{1}{T}\sum_{t=1}^{T} \mathbb{E}_{p(\mathbf{X}_t | \mathbf{X}_T)} \left[\Big\|\Big(\sigma_\theta(\mathbf{X}_t, t)^{1/2} - \mathrm{Chol}(\sigma_t(\mathbf{X}_t))\Big)\Big\|_2^2\right], \tag{50}$$

$$L_2(\theta) = \mathop{\mathbb{E}}_{(\boldsymbol{X},\boldsymbol{y})\sim \boldsymbol{D}} \mathop{\mathbb{E}}_{\boldsymbol{X}_0 \sim q_\theta(\boldsymbol{X}_0 | \boldsymbol{X}_T)} \Big[\mathrm{loss}\Big(\boldsymbol{X}_0, \boldsymbol{y}\Big)\Big], \tag{51}$$

where $\mathrm{Chol}(\cdot)$ denotes the Cholesky-like factorization.

## A.6 ADDITIONAL EXPERIMENTS

### A.6.1 LARGE-SCALE EXPERIMENT

Table 6: Comparison of average performance achieved a pre-trained ViT-B-16 model (fine-tuned on CIFAR-10 from ImageNet) and its reconfigured variant produced by DIRECTOR.

| Method | #params | ACC/MCC ↑ | AURC ↓ | AUROC ↑ | FPR95 ↓ | ECE ↓ | NLL ↓ | Brier ↓ |
|--------|---------|-----------|--------|---------|---------|-------|-------|---------|
| ViT-B-16 | 86M | **97.880** | **0.141** | **95.624** | **19.340** | 18.021 | 2.564 | 6.843 |
| DIRECTOR | 50M | 97.170 | 2.120 | 94.881 | 28.622 | **1.611** | **1.122** | **4.762** |

To stress test DIRECTOR, we further evaluate its scalability on a larger-scale experiment which requires reconfiguring a pre-trained ViT-B-16 model (pre-trained on the large-scale ImageNet data and fine-tuned on CIFAR-10 data). The reported results in Table 6 show that DIRECTOR 's reconfigured ViT-B-16 achieves substantially better uncertainty calibration with an ECE of 1.611 compared to 18.021 of the original ViT-B-16. DIRECTOR also maintains competitive predictive performance (97.17% vs. 97.88%) while requiring significantly fewer parameters (50M vs. 86M). These findings highlight the ability of DIRECTOR to deliver reliable uncertainty quantification even in large and complex transformer architectures, pre-trained on sophisticated and large-scale data.

Table 7: Performance comparison of Deep Ensembles for in-distribution classification on CIFAR-10 and IMDB. KEP-$k/n$ denotes a pre-trained transformer using GP-reparameterized architecture (KEP (Chen et al., 2024c)) for the last $k$ attention blocks and standard MHSA for the remaining blocks. **Bold** indicates the better performance in each pairwise comparison between a baseline ensemble (ViT, Transformer, KEP) and diffusion-based reconfigured KEP (Chen et al., 2024c) produced by DIRECTOR ensemble.

| Dataset | Method | ACC ↑ | AURC ↓ | AUROC ↑ | FPR95 ↓ | ECE ↓ | NLL ↓ | Brier ↓ |
|---|---|---|---|---|---|---|---|---|
| CIFAR-10 | ViT | 87.21 | 2.52 | 88.84 | 58.41 | 2.70 | 5.14 | 19.01 |
| | DIRECTOR | **88.82** | **1.91** | **90.09** | **52.77** | **1.72** | **3.69** | **16.38** |
| | KEP-1/7 | 87.48 | 2.33 | 89.43 | 56.39 | 2.30 | 4.46 | 18.29 |
| | DIRECTOR | **89.32** | **1.78** | **90.12** | **54.31** | **1.67** | **3.50** | **15.76** |
| | KEP-2/7 | 87.09 | 2.41 | 89.48 | 54.07 | 2.71 | 4.48 | 18.69 |
| | DIRECTOR | **89.20** | **1.79** | **90.33** | **51.85** | **1.86** | **3.52** | **15.73** |
| | KEP-7/7 | 83.97 | 3.77 | 86.90 | 62.38 | 3.58 | 5.04 | 23.20 |
| | DIRECTOR | **89.73** | **1.70** | **90.17** | **54.72** | **1.62** | **3.49** | **15.27** |
| IMDB | Transformer | 87.31 | **3.55** | **82.92** | 71.45 | 2.36 | 3.04 | 18.44 |
| | DIRECTOR | **87.45** | 3.56 | 82.83 | **70.68** | **1.97** | **3.00** | **18.25** |
| | KEP-1/5 | 87.44 | 3.63 | 82.34 | 72.21 | 1.56 | 3.01 | 18.39 |
| | DIRECTOR | **87.85** | **3.44** | **83.20** | **71.10** | **1.25** | **2.89** | **17.62** |
| | KEP-2/5 | 87.77 | 3.51 | 82.44 | 73.14 | 2.85 | 3.04 | 18.18 |
| | DIRECTOR | **87.88** | **3.43** | **82.96** | **72.29** | **1.24** | **2.91** | **17.70** |
| | KEP-5/5 | 86.49 | 4.16 | 81.54 | 74.07 | 2.63 | 3.28 | 19.72 |
| | DIRECTOR | **87.66** | **3.43** | **82.95** | **72.13** | **2.07** | **2.95** | **18.05** |

### A.6.2 DEEP ENSEMBLES RESULTS

We further evaluate the performance of Deep Ensembles (Lakshminarayanan et al., 2017), a simple yet effective method for uncertainty estimation, when combined with pretrained models and DIRECTOR, as reported in Table 7. The results show that DIRECTOR consistently outperforms pretrained models across both CIFAR-10 and IMDB. On CIFAR-10, DIRECTOR achieves notable improvements in predictive accuracy (up to +5.8% over KEP-7/7), while also delivering stronger uncertainty calibration (lower ECE and NLL) and robustness (AURC, AUROC, Brier score). On IMDB, although the baseline Transformer ensemble already achieves strong results, DIRECTOR further improves calibration (ECE, NLL, Brier) and robustness metrics, with the exception of AURC and AUROC where the Transformer baseline performs slightly better. Overall, these findings demonstrate that DIRECTOR works effectively to ensemble settings, providing consistent benefits across architectures, datasets, and evaluation metrics.

### A.6.3 IN-DISTRIBUTION CLASSIFICATION

We consolidate the experimental results from Table 1 and Table 2, and further include additional findings based on our alignment with KEP-2/7 (vision) and KEP-2/5 (language) configurations, presented in Table 8. DIRECTOR demonstrates state-of-the-art performance across nearly all scenarios, achieving the **highest** score in 23 out of 28 settings and ranking **second** in another 23 out of 28. These results highlight the substantial advantage of DIRECTOR over existing baselines, in terms of both predictive accuracy and uncertainty quantification.

Table 8: Performance comparison for in-distribution classification across four tasks. KEP-$k/n$ denotes a pre-trained transformer using GP-reparameterized architecture (KEP (Chen et al., 2024c)) for the last $k$ attention blocks and standard MHSA for the remaining blocks. **Bold** indicates the better performance in each pairwise comparison between a pre-trained model (ViT, Transformer, KEP) and diffusion-based reconfigured KEP (Chen et al., 2024c) produced by DIRECTOR . **Blue** marks the best result across all baselines for a dataset, and **brown** denotes the second-best.

| Dataset | Method | ACC/MCC↑ | AURC↓ | AUROC↑ | FPR95↓ | ECE↓ | NLL↓ | Brier↓ |
|---|---|---|---|---|---|---|---|---|
| CIFAR-10 | TS | 83.84±0.09 | 3.88±0.10 | 86.82±0.37 | 65.99±1.94 | 9.22±0.36 | 6.58±0.16 | 25.50±0.13 |
| | MCD | 84.06±0.23 | 8.65±0.03 | 86.51±0.32 | 66.15±0.60 | 9.47±0.16 | 8.36±0.32 | 25.45±0.29 |
| | KFLLA | 83.84±0.10 | 3.91±0.11 | 86.71±0.45 | 65.44±1.58 | 8.18±0.80 | 6.09±0.40 | 24.98±0.59 |
| | SV-DKL | 83.23±0.17 | 4.39±0.18 | 85.94±0.36 | 66.96±1.30 | 11.64±0.81 | 9.85±1.09 | 27.97±0.77 |
| | SGPA | 75.59±3.63 | 8.41±2.37 | 82.65±1.71 | 71.78±2.73 | 1.92±0.55 | 7.11±0.95 | 33.98±4.57 |
| | ViT | 83.84±0.09 | 4.05±0.11 | 86.42±0.37 | 67.13±1.98 | 12.51±0.20 | 10.91±0.39 | 28.03±0.15 |
| | DIRECTOR | **85.67**±0.88 | **3.10**±0.45 | **87.90**±1.04 | **61.92**±1.85 | **9.67**±0.62 | **6.94**±0.52 | **23.54**±1.46 |
| | KEP-1/7 | 84.52±0.25 | 3.52±0.11 | 87.52±0.27 | 65.14±1.27 | 10.93±0.26 | 8.21±0.15 | 25.80±0.40 |
| | DIRECTOR | **86.51**±0.57 | **2.78**±0.16 | **88.28**±0.16 | **61.85**±1.93 | **8.61**±0.49 | **6.11**±0.27 | **21.90**±0.82 |
| | KEP-2/7 | 84.32±0.75 | 3.65±0.22 | 87.17±0.35 | 65.30±1.35 | 11.11±0.46 | 8.43±0.26 | 26.20±1.08 |
| | DIRECTOR | **86.62**±0.18 | **2.70**±0.07 | **88.52**±0.36 | **60.70**±2.31 | **8.54**±0.14 | **5.96**±0.14 | **21.59**±0.25 |
| | KEP-7/7 | 82.68±0.12 | 4.46±0.05 | 85.71±0.34 | 66.90±1.78 | 6.95±0.36 | 5.89±0.11 | 25.90±0.23 |
| | DIRECTOR | **87.06**±0.18 | **2.57**±0.11 | **88.60**±0.49 | 61.74±0.52 | 8.42±0.15 | 5.96±0.18 | **21.11**±0.32 |
| CIFAR-100 | TS | 52.94±0.63 | 22.34±0.61 | 82.29±0.48 | 71.65±1.98 | 17.06±0.42 | 21.57±0.52 | 64.77±0.95 |
| | MCD | 53.49±0.62 | 22.24±0.56 | 81.60±0.19 | 73.02±0.51 | 25.93±0.37 | 29.24±0.73 | 70.02±0.92 |
| | KFLLA | 52.27±0.86 | 23.96±0.78 | 81.30±0.48 | 71.42±1.92 | 18.52±5.40 | 20.89±0.57 | 66.51±1.66 |
| | SV-DKL | 51.03±0.60 | 24.38±0.43 | 81.32±0.50 | 73.99±1.40 | 25.46±0.72 | 28.93±0.66 | 71.90±0.74 |
| | SGPA | 52.77±0.52 | 22.84±0.52 | 81.65±0.36 | 72.02±1.74 | 10.33±2.25 | 19.10±0.57 | 62.08±1.04 |
| | ViT | 52.94±0.63 | 22.88±0.64 | 81.07±0.56 | 75.96±2.39 | 30.73±0.61 | 32.71±0.91 | 74.72±1.20 |
| | DIRECTOR | **57.79**±0.57 | **18.58**±0.37 | **82.60**±0.11 | **71.50**±1.42 | **22.31**±0.49 | **22.16**±0.36 | **62.57**±0.85 |
| | KEP-1/7 | 55.74±0.77 | 20.20±0.64 | 82.10±0.20 | 73.82±0.92 | 27.07±0.71 | 27.45±0.62 | 68.54±1.18 |
| | DIRECTOR | **58.78**±1.52 | **17.63**±1.08 | **82.99**±0.41 | **71.40**±1.24 | **21.17**±1.03 | **21.15**±0.93 | **60.96**±1.73 |
| | KEP-2/7 | 55.82±1.12 | 20.14±0.94 | 82.16±0.32 | 73.34±1.05 | 27.37±0.55 | 27.74±0.51 | 68.65±1.42 |
| | DIRECTOR | **59.45**±1.06 | **17.10**±0.80 | **83.27**±0.33 | **69.94**±1.33 | **21.44**±0.86 | **20.85**±0.49 | **60.15**±1.16 |
| | KEP-7/7 | 57.06±0.56 | 19.38±0.60 | 82.02±0.39 | 72.78±0.69 | 21.31±3.85 | 22.41±2.29 | 63.07±2.11 |
| | DIRECTOR | **60.85**±2.98 | **16.35**±2.34 | **82.91**±0.67 | **71.46**±1.43 | 21.43±2.09 | **20.55**±2.42 | **58.91**±4.63 |
| IMDB | TS | 85.59±0.50 | 4.73±0.32 | 80.75±0.55 | 75.45±1.10 | 2.91±1.51 | 3.41±0.13 | 21.04±0.74 |
| | MCD | 85.96±0.42 | 4.40±0.24 | 81.40±0.55 | 74.79±0.88 | 4.18±2.03 | 3.47±0.23 | 20.72±0.82 |
| | KFLLA | 85.59±0.50 | 4.71±0.30 | 80.82±0.48 | 75.45±1.11 | 5.84±2.21 | 6.93±0.00 | 21.86±1.19 |
| | SV-DKL | 85.69±0.66 | 5.58±0.79 | 78.54±2.20 | 75.32±0.84 | 8.52±1.57 | 4.49±0.65 | 23.10±1.66 |
| | SGPA | 85.39±0.36 | 4.96±0.49 | 80.04±1.14 | 76.44±0.96 | 6.04±1.71 | 3.96±0.50 | 22.19±1.06 |
| | Transformer | 85.59±0.50 | 4.73±0.32 | 80.75±0.55 | 75.45±1.10 | 6.96±2.05 | 3.95±0.44 | 22.28±1.26 |
| | DIRECTOR | **86.07**±0.61 | **4.57**±0.39 | **80.84**±0.67 | **74.24**±0.87 | **5.40**±1.90 | **3.60**±0.34 | **21.08**±1.32 |
| | KEP-1/5 | 85.76±0.71 | 4.54±0.42 | 81.02±0.70 | 74.87±0.87 | 5.51±2.94 | 3.79±0.51 | 21.62±1.42 |
| | DIRECTOR | **87.13**±0.19 | **4.07**±0.30 | **81.55**±0.62 | **73.35**±0.14 | **3.16**±2.54 | **3.24**±0.31 | **19.31**±0.97 |
| | KEP-2/5 | 86.52±0.72 | 4.18±0.37 | 81.54±0.54 | 73.82±1.68 | 5.72±1.20 | 3.58±0.29 | 20.51±1.16 |
| | DIRECTOR | **86.95**±0.19 | **4.05**±0.30 | **81.64**±0.85 | **73.46**±1.34 | **3.46**±1.40 | **3.23**±0.13 | **19.42**±0.49 |
| | KEP-5/5 | 84.57±0.81 | 5.48±0.60 | 79.32±1.25 | 77.03±1.48 | 7.83±3.28 | 5.17±2.13 | 24.23±2.56 |
| | DIRECTOR | **85.74**±0.34 | **4.58**±0.23 | **80.95**±0.42 | **75.08**±1.15 | **2.33**±1.54 | **3.36**±0.12 | **20.70**±0.65 |
| CoLA | TS | 29.92±1.17 | 20.84±1.23 | 64.31±1.44 | 89.93±2.95 | 23.22±2.99 | 11.04±1.91 | 51.70±3.42 |
| | MCD | 30.04±1.02 | 20.66±1.11 | 64.53±1.00 | 89.51±1.35 | 24.96±1.79 | 17.83±3.61 | 53.52±2.59 |
| | KFLLA | 29.89±1.14 | 20.82±1.26 | 64.22±1.46 | 89.87±3.24 | 24.36±2.25 | 12.16±1.49 | 52.80±2.85 |
| | SV-DKL | 30.07±1.41 | 22.76±2.28 | 61.98±3.09 | 89.00±2.55 | 25.71±1.60 | 17.96±3.26 | 54.40±2.13 |
| | SGPA | 31.53±2.05 | 20.44±2.60 | 64.34±1.95 | 90.79±0.87 | 26.22±1.51 | 28.65±7.23 | 54.08±2.44 |
| | Transformer | 29.92±1.17 | 20.80±1.21 | 64.22±1.46 | 90.01±2.84 | 26.44±1.90 | 19.66±4.18 | 55.09±2.68 |
| | DIRECTOR | **31.85**±2.46 | **19.74**±1.62 | **64.52**±1.71 | **89.52**±3.61 | **23.94**±0.49 | **14.29**±3.01 | **50.82**±1.19 |
| | KEP-1/5 | 30.86±2.03 | 19.86±2.04 | 65.18±1.83 | 89.10±2.60 | 24.98±2.08 | 16.28±4.35 | 52.86±2.88 |
| | DIRECTOR | 31.84±1.88 | **19.42**±1.83 | **65.54**±1.78 | 90.37±0.89 | **13.77**±6.58 | **8.26**±3.75 | **43.54**±3.65 |
| | KEP-2/5 | 30.73±2.29 | 20.36±1.40 | 64.03±1.21 | 90.23±1.86 | 21.78±6.62 | 15.52±7.74 | 50.14±5.92 |
| | DIRECTOR | **33.03**±1.36 | **19.06**±3.05 | **64.78**±2.31 | **89.79**±2.65 | **11.45**±4.78 | **6.72**±0.98 | **41.64**±3.58 |
| | KEP-5/5 | 29.28±1.21 | 20.82±1.98 | 64.47±0.90 | 89.65±1.04 | 18.95±5.66 | 11.83±7.96 | 48.65±5.84 |
| | DIRECTOR | **32.05**±1.56 | **18.69**±1.39 | **64.71**±1.37 | **89.53**±1.67 | **12.88**±5.74 | **7.68**±1.70 | **42.20**±2.50 |

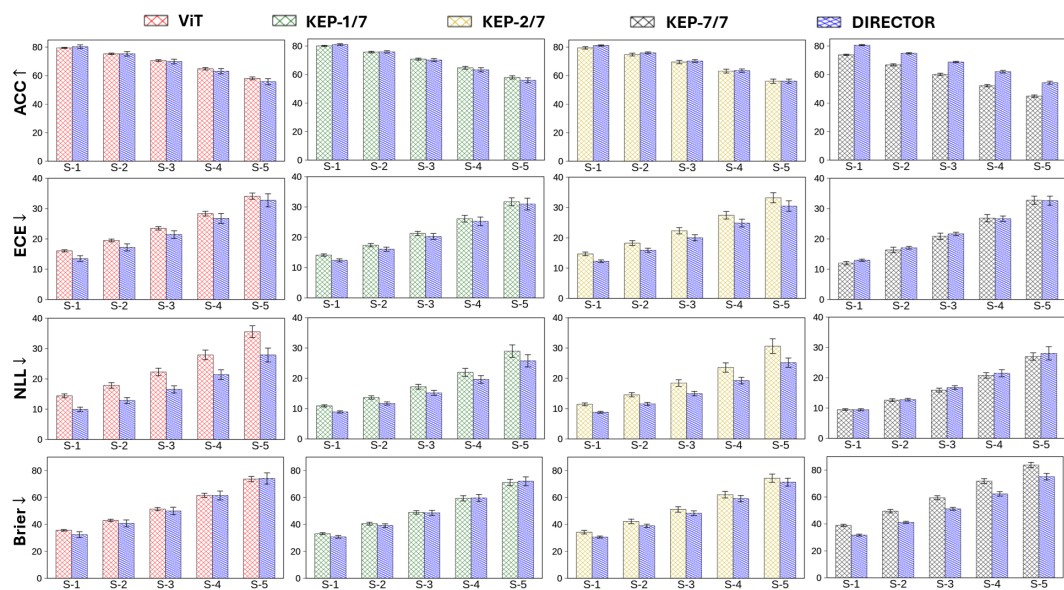

Figure 3: Calibration comparison of pre-trained models with their corresponding diffusion-based reconfigured produced by DIRECTOR on CIFAR-10-C over 5 severity levels of corruption. The notation S-k represents the severity level $k$. DIRECTOR achieves competitive accuracy and outperforms pre-trained models in most calibration metrics.

Table 9: Performance comparison on CIFAR-10-C, with 15 corruptions across five severity levels over five trials.

| Method | ACC ↑ | AURC ↓ | AUROC ↑ | FPR95 ↓ | ECE ↓ | NLL ↓ | Brier ↓ |
|---|---|---|---|---|---|---|---|
| MCD | 69.73±0.36 | 14.46±0.22 | 79.11±0.11 | 76.96±0.20 | 19.58±0.23 | 18.48±0.67 | 48.70±0.47 |
| KFLLA | 69.64±0.43 | 14.46±0.41 | 79.27±0.23 | 76.34±0.33 | 17.46±1.18 | 12.75±0.80 | 47.18±1.06 |
| SVDKL | 68.88±0.36 | 15.43±0.56 | 78.56±0.50 | 77.22±0.55 | 22.67±1.18 | 21.02±2.57 | 52.29±1.35 |
| SGPA | 57.73±1.25 | 24.82±1.23 | 74.81±0.65 | 80.92±0.54 | 12.47±2.11 | 13.60±0.48 | 57.91±1.39 |
| ViT | **69.67**±0.34 | **14.66**±0.27 | 78.92±0.16 | 77.74±0.28 | 24.30±0.31 | 23.59±1.00 | 53.07±0.59 |
| DIRECTOR | 68.89±1.44 | 15.31±1.30 | **79.17**±0.86 | **77.10**±1.00 | **22.32**±1.09 | **17.71**±1.02 | **51.77**±2.39 |
| KEP-1/7 | **69.87**±0.45 | **14.30**±0.50 | 79.49±0.38 | 76.95±0.43 | 22.12±0.47 | 18.54±0.63 | 50.65±0.90 |
| DIRECTOR | 69.29±0.66 | 15.00±0.69 | **79.69**±0.27 | **76.38**±0.40 | **20.98**±0.65 | **16.23**±0.60 | **50.07**±1.20 |
| KEP-2/7 | 68.63±0.75 | 15.46±0.66 | 78.90±0.37 | 77.57±0.42 | 23.18±0.63 | 19.72±0.76 | 52.82±1.30 |
| DIRECTOR | **69.41**±0.55 | **14.95**±0.60 | **79.73**±0.42 | **76.29**±0.73 | **20.71**±0.63 | **15.94**±0.44 | **49.71**±1.10 |
| KEP-7/7 | 59.57±0.30 | 23.77±0.40 | 75.56±0.30 | 80.47±0.27 | **21.78**±0.59 | **17.17**±0.39 | 60.67±0.72 |
| DIRECTOR | **68.12**±0.26 | **16.45**±0.28 | **79.08**±0.22 | **76.73**±0.29 | 22.19±0.19 | 17.70±0.17 | **52.27**±0.35 |

### A.6.4 DISTRIBUTION SHIFT ROBUSTNESS

Additional experiments evaluating the distributional robustness of DIRECTOR are presented in Tables 9,10, and Figure3. These results include evaluations of DIRECTOR aligned with KEP-2/5 and KEP-2/7 configurations. Notably, the TS baseline is excluded from these comparisons, as it is specifically tailored for in-distribution tasks.

Figure 3 illustrates a calibration comparison between pre-trained models (ViT, KEP) and DIRECTOR on the CIFAR-10-C dataset, across five corruption severity levels averaged over 15 corruption types. DIRECTOR demonstrates competitive, and in many cases superior, predictive accuracy, particularly when aligned with KEP-7/7. In terms of uncertainty quantification, DIRECTOR exhibits significantly improved calibration, achieving lower values in ECE, NLL, and Brier score compared to pre-trained baselines.

Table 9 summarizes performance averaged over the 15 corruptions and five severity levels on CIFAR-10-C. While maintaining competitive predictive accuracy with ViT and KEP, DIRECTOR substan-

Table 10: Performance comparison on CoLA OOD over five trials.

| Method | MCC ↑ | AURC ↓ | AUROC ↑ | FPR95 ↓ | ECE ↓ | NLL ↓ | Brier ↓ |
|---|---|---|---|---|---|---|---|
| MC Dropout | 18.54±4.14 | 25.85±1.03 | 63.31±2.18 | 90.17±2.95 | 32.02±2.56 | 23.84±5.19 | 65.50±4.46 |
| KFLLA | 18.43±3.55 | 25.89±0.99 | 63.31±1.65 | 90.50±3.25 | 29.94±2.86 | 14.64±1.87 | 62.72±4.37 |
| SVDKL | 19.32±3.57 | 27.58±2.51 | 60.65±1.41 | 89.97±2.67 | 30.97±3.10 | 21.23±3.67 | 64.07±4.62 |
| SGPA | 19.34±6.23 | 27.48±2.55 | 61.68±2.29 | 90.14±2.30 | 31.15±1.66 | 35.18±8.59 | 63.62±3.65 |
| Transformer | 18.43±3.55 | 25.85±1.00 | **63.38**±1.69 | 90.39±3.27 | 31.99±2.70 | 23.82±5.16 | 65.50±4.32 |
| DIRECTOR | **23.06**±4.69 | **25.61**±2.30 | 61.32±3.21 | **88.23**±3.89 | **28.72**±1.87 | **17.18**±3.75 | **59.91**±2.20 |
| KEP-1/5 | 19.44±1.94 | 25.21±1.53 | **63.65**±3.19 | **87.35**±2.43 | 30.33±1.48 | 19.67±4.82 | 61.97±2.25 |
| DIRECTOR | **22.10**±5.49 | **24.91**±1.80 | 62.23±2.98 | 91.55±3.19 | **17.50**±7.52 | **9.34**±4.50 | **49.92**±5.80 |
| KEP-2/5 | **20.38**±5.22 | 24.39±2.05 | 63.60±1.71 | 90.75±2.62 | 26.04±7.51 | 17.94±9.12 | 57.77±8.04 |
| DIRECTOR | 20.11±2.48 | **23.23**±2.36 | **63.62**±0.91 | **89.81**±1.37 | **15.67**±5.90 | **7.64**±1.45 | **48.23**±5.01 |
| KEP-5/5 | **21.14**±3.48 | 24.06±2.27 | 63.33±1.36 | 90.46±2.20 | 23.27±6.61 | 14.25±10.66 | 55.12±7.78 |
| DIRECTOR | 20.20±5.46 | **21.97**±1.72 | **65.12**±1.63 | 90.17±3.89 | **17.49**±5.93 | **8.61**±1.99 | **48.55**±3.67 |

Table 11: Ablation on loss weighting on CoLA when diffusion-based reconfiguring KEP-5/5 with DIRECTOR. We report mean ± std across 5 seeds. The configuration $(\lambda_{\text{mean}}, \lambda_{\text{Chol}}, \lambda_{\text{NLL}}) = (0.5, 0.2, 0.3)$ (gray) is chosen as it balances predictive performance (MCC) with uncertainty calibration. This setting is applied consistently for all diffusion-based reconfigured KEP (Chen et al., 2024c) produced by DIRECTOR .

| Method | $\lambda_{\text{mean}}$ | $\lambda_{\text{Chol}}$ | $\lambda_{\text{NLL}}$ | MCC ↑ | ECE ↓ | NLL ↓ |
|---|---|---|---|---|---|---|
| Transformers | – | – | – | 29.92±1.17 | 26.44±1.90 | 19.66 ± 4.18 |
| DIRECTOR | 0.9 | 0.05 | 0.05 | 29.98±3.45 | 5.96±4.60 | 5.86±0.49 |
|  | 0.6 | 0.1 | 0.3 | 33.18±1.53 | 18.36±2.29 | 10.09±2.38 |
|  | 0.5 | 0.3 | 0.2 | 32.26±2.87 | 15.27±9.06 | 9.51±2.70 |
|  | 0.5 | 0.25 | 0.25 | 32.96±1.85 | 12.27±8.57 | 8.23±2.71 |
|  | 0.5 | 0.2 | 0.3 | 31.81±2.30 | 12.65±6.04 | 7.69±1.70 |
|  | 0.3 | 0.2 | 0.5 | 31.86±2.79 | 18.64±8.43 | 13.77±7.32 |
|  | 0.3 | 0.1 | 0.6 | 30.21±2.32 | 16.50±8.60 | 10.74±5.80 |
|  | 0.25 | 0.25 | 0.5 | 33.38±2.08 | 22.26±3.72 | 15.10±4.85 |
|  | 0.05 | 0.05 | 0.9 | 30.74±1.44 | 15.50±6.34 | 9.52±4.12 |

tially outperforms them in calibration metrics. When aligned with KEP-2/7, our best-performing configuration of DIRECTOR achieves both competitive accuracy and slightly improved calibration compared to other baselines. Additionally, DIRECTOR can be enhanced by integrating post-training baselines such as MCD or aligning with attention-modified methods like SGPA to further improve calibration. However, due to computational constraints, we leave these extensions for future investigation.

Finally, Table 10 compares DIRECTOR against pre-trained models (Transformer and KEP) and other baselines on the CoLA OOD dataset. DIRECTOR achieves the highest MCC score, outperforming all baselines, and shows significant gains in both calibration and failure prediction metrics.

### A.6.5 EFFECT OF VARYING LOSS WEIGHTS

We perform extensive ablation studies on the loss weight configurations to investigate their impact on DIRECTOR's performance on both the CoLA and CIFAR-10 datasets. Specifically, we examine how varying the weights assigned to the mean matching term ($\lambda_{\text{mean}}$), the Cholesky-like factor alignment ($\lambda_{\text{Chol}}$), and the performance-aware loss ($\lambda_{\text{NLL}}$) affects both predictive accuracy and uncertainty calibration. When applying diffusion-based reconfiguration with DIRECTOR to KEP-5/5 on CoLA (see Table 11), we observe distinct trends: excessively high $\lambda_{\text{mean}}$, as in the configuration $(0.9, 0.05, 0.05)$, produces very strong calibration while maintaining decent predictive performance. Conversely, assigning too much weight to $\lambda_{\text{NLL}}$, as in $(0.05, 0.05, 0.9)$, prioritizes accuracy at the expense of calibration, leading to more confident yet less well-calibrated predictions.

Table 12: Ablation on loss weighting on CIFAR-10 (single run) when reconfiguring ViT with DIRECTOR. We report test accuracy (ACC), Expected Calibration Error (ECE), and Negative Log-Likelihood (NLL). The configuration $(\lambda_{\text{mean}}, \lambda_{\text{NLL}}) = (0.8, 0.2)$ (gray) is selected as it achieves the best balance between accuracy and calibration, and is used for all diffusion-based reconfigured ViT produced by DIRECTOR .

| Method | $\lambda_{\text{mean}}$ | $\lambda_{\text{NLL}}$ | ACC ↑ | AURC ↓ | AUROC ↑ | FPR95 ↓ | ECE ↓ | NLL ↓ | Brier ↓ |
|---|---|---|---|---|---|---|---|---|---|
| ViT | – | – | 84.22 | 3.90 | 86.62 | 65.34 | 12.39 | 10.67 | 27.52 |
| DIRECTOR | 0.0 | 1.0 | 84.03 | 4.01 | 85.97 | 67.81 | 12.12 | 9.69 | 27.49 |
| | 0.1 | 0.9 | 85.24 | 3.26 | 87.54 | 62.87 | 10.97 | 8.51 | 24.99 |
| | 0.2 | 0.8 | 85.67 | 3.03 | 88.16 | 61.83 | 10.52 | 7.75 | 24.11 |
| | 0.3 | 0.7 | 86.08 | 3.06 | 87.57 | 59.91 | 10.24 | 7.86 | 23.44 |
| | 0.4 | 0.6 | 86.05 | 2.91 | 88.51 | 62.22 | 10.25 | 7.61 | 23.45 |
| | 0.5 | 0.5 | 86.18 | 2.86 | 88.21 | 63.89 | 9.97 | 7.41 | 23.45 |
| | 0.6 | 0.4 | 86.60 | 2.83 | 87.81 | 64.93 | 9.50 | 6.98 | 22.58 |
| | 0.7 | 0.3 | 86.66 | 2.73 | 88.39 | 61.02 | 9.18 | 6.74 | 22.05 |
| | 0.8 | 0.2 | **87.16** | **2.52** | **88.92** | **57.94** | **8.54** | **6.06** | **20.99** |
| | 0.9 | 0.1 | 85.09 | 3.54 | 86.73 | 64.19 | 10.15 | 7.37 | 24.79 |

Intermediate configurations, such as $(0.5, 0.2, 0.3)$, provide a balanced trade-off, achieving high predictive performance while significantly improving calibration. Based on these insights, we adopt $(0.5, 0.2, 0.3)$ consistently for all diffusion-based reconfigured KEP produced by DIRECTOR, as it offers the most reliable combination of accuracy and calibrated uncertainty.

We also study the effect of varying the weights for the mean matching term ($\lambda_{\text{mean}}$) and the performance-aware loss ($\lambda_{\text{NLL}}$) while setting $\lambda_{\text{Chol}} = 0$ during diffusion-based reconfiguring ViT with DIRECTOR on CIFAR-10 (Table 12). When $\lambda_{\text{mean}}$ is excluded (e.g., $(0.0, 1.0)$), the NLL term dominates, which may slightly improve calibration but leads to reduced predictive performance. In contrast, including both mean matching ($\lambda_{\text{mean}} \neq 0$) and performance-aware loss consistently enhances accuracy and uncertainty calibration. Notably, the configuration $(0.8, 0.2)$ achieves the best overall balance, with the highest accuracy (87.16) and lowest ECE (8.54). Consequently, we adopt $(0.8, 0.2)$ for all diffusion-based reconfigured ViT produced by DIRECTOR.

### A.6.6 TRAINING CONFIGURATION AND LEARNING DYNAMICS

In the first step, we reproduce the ViT and KEP-SVGP architectures using the previously described settings. We then reconfigure each pre-trained model using a DiT-based unified transition model, which takes as input the pair $(X_t, t)$ and outputs the feature representation at diffusion step $t$. The unified transition model (DiT) conditions on the time step using adaptive LayerNorm-zero (AdaLN-Zero), and we use the default DiT configuration, with the model dimension $d_{\text{model}}$ adjusted according to the corresponding pretrained backbone. We show the training loss curves and validation accuracy over epochs for ViT and KEP-7/7 on CIFAR-10 in Fig. 4. All models are trained under the same experimental settings as described in Chen et al. (2024c), and the curves indicate that the training loss has converged for both architectures. For reproducibility, the training hyperparameters used for the unified transition model are summarized in Table 13.

Table 13: Hyperparameters for unified transition model training.

| Hyperparameters | Settings |
|---|---|
| $d_{\text{model}}$ | 384 (CV), 256 (CoLA), 128 (IMDB) |
| Depth of DiT | 1 |
| Epochs | 100 (CV), 50 (NLP) |
| Batch size | 128 (CV), 32 (NLP) |
| Dropout | 0.1 |
| Optimizer | Adam |
| Learning rate | 1e-3 (CV), 5e-3 (NLP) |
| Adam $\beta$ | (0.9, 0.999) |
| Weight decay | 1e-5 |
| Scheduler | CosineAnnealingLR |
| Cosine cycle epochs | 50 |
| Minimal learning rate | 1e-5 |
| Warmup Epochs | 5 |

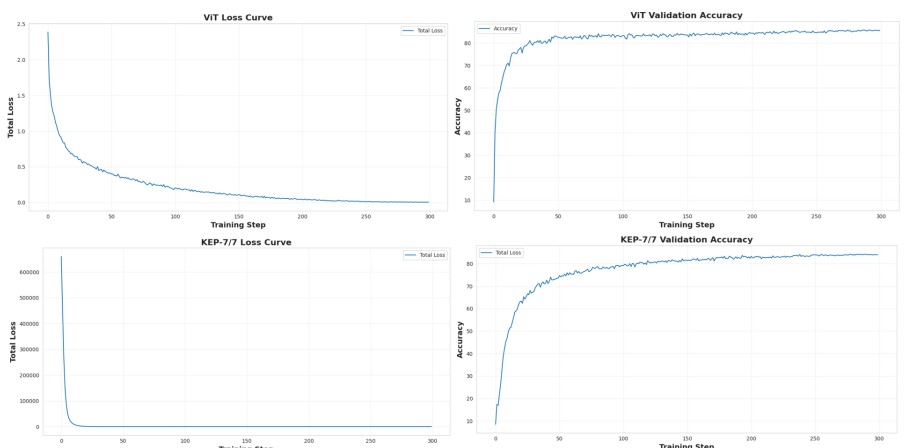

Figure 4: Loss curves and validation accuracy across epochs for ViT and KEP-7/7.

