# OpenReview forum: "Diffusion-Inspired Reconfiguration of Transformers for Uncertainty Quantification"
_ICLR.cc/2026/Conference — Submitted to ICLR 2026_

### Official Review · Reviewer_UFSu · 2025-10-21

**Soundness:** 3
**Presentation:** 4
**Contribution:** 4
**Rating:** 6
**Confidence:** 2

**Summary:**

The paper follows a recent line of research that views transformers as diffusion processes in the embedding space. The paper proposes to record the embedding state after each multi-head attention block and fit a diffusion model to fit the trajectory of embeddings from input (t=T) to final-layer embedding (t=0). This is a subtle change in how the diffusion process is modeled and constructed from the pretrained transformer in previous works. The authors report to match or outperform the transformers they distill from across the line. The experiments are on computer vision and NLP on a small ~3M scale.

**Strengths:**

In order of magnitude:

1. The relevance and novelty of this line of research is intruiging. However, I cannot judge the exact novelty over the previous three papers from Chen et al., which it seems to iterate on. I hope that the other reviewers have read these previous papers and can comment on how strong the novelty of the proposed changes is.
2. The proposed method seems to match or outperform the baselines in terms of accuracy while improving ECE/NLL/Brier score.
3. The evaluation suite is wide in terms of metrics and baseline (variational) methods and consistently reports standard errors.
4. The code is provided and gives all exact commands to reproduce the results of the paper.
5. The writing is very clear and the mathematical notation makes it easy to follow the details.

**Weaknesses:**

In order of magnitude:

1. I am somewhat concerned with the stability / robustness to deeper transformers. Currently, all experiments are on very small (<= 6M) and very shallow (5 to 7 layer) transformers, whereas more modern models tend to have 4x the layers and about 1000x the parameters. Besides the transfer to 1000x larger models (which often fails), I am most concerned that the probability trajectories might not be distilled as well on deeper architectures (= more diffusion steps). It would be great if you could give at least single points of evidence here for NLP transformers, like a small Qwen 2.5 model (on top of the ViT-B-16 experiment in Appendix A.6.1)
2. Compute permitting (duly noted that “all experiments are conducted on a single NVIDIA L40 GPU”), it would be great to evaluate on bigger test datasets. Currently, the analysis rests on 2x 10k test samples in computer vision and 25k + 2k test samples in NLP. E.g., in computer vision, there is ImageNet instead of CIFAR, and maybe DINO or AIM-like setups, and in language there is the lm eval harness.
3. The baseline transformers appear to have been trained on very little compute. I am not sure if the performance is converged, particularly because the standard error is within the performance delta (and so the standard deviation is even larger). It would be great to distill from pretrained models from the literature (to make sure they are converged). It would also be great to detail your train pipeline, i.e., what has been distilled from what for how many epochs, etc. This can prevent seeing improved results simply because the distilled method has effectively been trained for longer.
4. It would be nice, but not high priority, to add a distillation baseline that distills a student transformer from the teacher transformer, using the same pipeline (= epochs, data, ...) you use for your method. This can make sure that the gains you observe are not just due to soft-label distillation, longer training, or any other maybe hidden difference that the distillation pipeline introduces.

### Smaller weaknesses that don’t influence my score and don’t need to be rebuttled, but would be nice to fix in the revised version

* It would be great to bring your deep ensemble comparisons from the appendix to the main paper in Table 2, especially because you claim to set a new SOTA on uncertainty quantification.
* I am not sure how you structured your experiments with the random seeds. If you pretrained five models, and then distilled them each, then you could do coupled comparisons to minimize the estimation variance of the delta your method brings.
* I would suggest to move the related works to the front, and to use it to elaborate a bit more on distilling diffusion trajectories from transformers, so basically the Chen et al. line of research

## Justification for the overall score

The paper is refreshing and works on bridging two important current techniques with attention to detail. I think it offers a nice perspective to the conference and should be accepted. However, I remain cautious of a too enthusiastic recommendation since all experiments are on very small scale and may not be relevant for larger modern models. I am happy to increase my score if the authors can resolve this doubt in the rebuttal period.

**Questions:**

1. Can you elaborate on which parts of your perspective come from the previous Chen et al. research line and where you differ? From lines 127-133 it seems like the main novelty is that you stabilize training by recording embeddings after each attention block, treating MLP and LN as part of what function the next-step GP models (as opposed to something different before)?
2. How are the baselines pretrained? E.g., how many epochs do you use, and have you considered using pretrained models from other researchers (which have the benefit of likely having converged in training)
3. Is there any benefits that the GPs give you in terms of disentangling aleatoric and epistemic uncertainty? Maybe you could launch a simple experiment, but this has lower priority compared to the other experiments above.

---

> ### Author Response · Authors · 2025-11-25
> **Response to Reviewer UFSu (Part 1/3)**
>
> We would like to thank the reviewer for positive rating with insightful feedbacks and we will address the remaining concerns as follows:
>
> > I am somewhat concerned with the stability / robustness to deeper transformers. Currently, all experiments are on very small (<= 6M) and very shallow (5 to 7 layer) transformers, whereas more modern models tend to have 4x the layers and about 1000x the parameters. Besides the transfer to 1000x larger models (which often fails), I am most concerned that the probability trajectories might not be distilled as well on deeper architectures (= more diffusion steps). It would be great if you could give at least single points of evidence here for NLP transformers, like a small Qwen 2.5 model (on top of the ViT-B-16 experiment in Appendix A.6.1)
>
> > Compute permitting (duly noted that “all experiments are conducted on a single NVIDIA L40 GPU”), it would be great to evaluate on bigger test datasets. Currently, the analysis rests on 2x 10k test samples in computer vision and 25k + 2k test samples in NLP. E.g., in computer vision, there is ImageNet instead of CIFAR, and maybe DINO or AIM-like setups, and in language there is the lm eval harness.
>
> We are currently conducting these experiments. Given our limited computational resources, we will report the results in a subsequent revision.
>
> > The baseline transformers appear to have been trained on very little compute. I am not sure if the performance is converged, particularly because the standard error is within the performance delta (and so the standard deviation is even larger). It would be great to distill from pretrained models from the literature (to make sure they are converged). It would also be great to detail your train pipeline, i.e., what has been distilled from what for how many epochs, etc. This can prevent seeing improved results simply because the distilled method has effectively been trained for longer.
> > How are the baselines pretrained? E.g., how many epochs do you use, and have you considered using pretrained models from other researchers (which have the benefit of likely having converged in training)
>
> We would like to clarify that the pretrained transformer models used in our experiments were generated following the same settings reported in KEP-SVGP [1] which does not release pretrained weights. To provide further transparency, we have added detailed descriptions of the training procedures for both the pretrained models and our method in the Appendix A.6.6. Additionally, we include the training loss curves of the pretrained models in Figure 4, which clearly demonstrate that these models have converged. This ensures that the observed improvements are not due to differences in training duration or convergence. Furthermore, for large-scale experiments, we also use publicly available pretrained models from other researchers (i.e., ViT-B-16 pretrained on ImageNet21k and fine-tuned on CIFAR-10: https://huggingface.co/aaraki/vit-base-patch16-224-in21k-finetuned-cifar10), as described in Appendix A.1.6.
>
> [1] Yingyi Chen, Qinghua Tao, Francesco Tonin, and Johan Suykens. Self-attention through kernel-eigen pair sparse variational gaussian processes. In Forty-first International Conference on Machine
> Learning, 2024b.

---

> ### Author Response · Authors · 2025-11-25
> **Response to Reviewer UFSu (Part 2/3)**
>
> > It would be nice, but not high priority, to add a distillation baseline that distills a student transformer from the teacher transformer, using the same pipeline (= epochs, data, ...) you use for your method. This can make sure that the gains you observe are not just due to soft-label distillation, longer training, or any other maybe hidden difference that the distillation pipeline introduces.
>
> As requested, we have run additional experiments to compare DIRECTOR with knowledge distillation (KD) [2]. The results are reported in table below. In particular, we evaluate KD using a pretrained 7-layer ViT as the teacher and a 3-layer ViT or the DiT parameterization of our transition probability as the student, trained with KD (temperature scaling = 3), while DIRECTOR use DiT and trained with our loss function. Although KD improves accuracy, its calibration metrics remain higher compared to our method, which achieves superior accuracy and calibration. Additionally, our propagation model is highly parameter-efficient, using only 2.7M parameters which is less than half of the original model's 6.27M parameters, resulting in improved memory efficiency.
>
> | Method   | Model         | #params | CIFAR-10 ACC ↑ | CIFAR-10 ECE ↓ | CIFAR-10 NLL ↓ | CIFAR-10 Brier ↓ | CIFAR-100 ACC ↑ | CIFAR-100 ECE ↓ | CIFAR-100 NLL ↓ | CIFAR-100 Brier ↓ |
> |----------|---------------|---------|----------------|----------------|----------------|-----------------|----------------|-----------------|-----------------|------------------|
> | Teacher  | ViT (7 layers)| 6.27M   | 83.84±0.09     | 12.51±0.20     | 10.91±0.39     | 28.03±0.15      | 52.94±0.63     | 30.73±0.61      | 32.71±0.91      | 74.72±1.20       |
> | Student  | ViT (3 layers)| 2.75M   | 82.20±0.80     | 10.92±1.53     | 7.90±0.80      | 28.58±1.20      | 54.86±0.65     | 30.12±0.39      | 31.25±0.53      | 72.17±0.78       |
> | Student  | DiT           | 2.70M   | 84.77±0.63     | 11.93±0.51     | 10.26±0.52     | 26.55±1.17      | 57.56±0.37     | 27.37±0.62      | 27.10±0.77      | 67.06±0.83       |
> | DIRECTOR | DiT           | 2.70M   | **85.67±0.88** | **9.67±0.62**  | **6.94±0.52**  | **23.54±1.46**  | **57.79±0.57** | **22.31±0.49**  | **22.16±0.36**  | **62.57±0.85**   |
>
> [2] Geoffrey Hinton, Oriol Vinyals, and Jeff Dean. Distilling the knowledge in a neural network, 2015.
>
> >Smaller weaknesses that don’t influence my score and don’t need to be rebuttled, but would be nice to fix in the revised version
>
> We would like to thank the reviewer for the detailed suggestions and will incorporate these improvements in the revised manuscript.
>
> > Can you elaborate on which parts of your perspective come from the previous Chen et al. research line and where you differ? From lines 127-133 it seems like the main novelty is that you stabilize training by recording embeddings after each attention block, treating MLP and LN as part of what function the next-step GP models (as opposed to something different before)?
>
> We would like to elaborate that prior works primarily focus on reparameterizing the attention mechanism using GP-based methods, and subsequently propagate uncertainty by simulating local representation uncertainty within each MHSA block. However, this propagation is disrupted by the interleaving of deterministic, point-estimated components such as MLPs and Layer Normalization (LN), as shown in Eq. (3) and the upper part of Figure 2. These components break the uncertainty flow and prevent a coherent probabilistic interpretation across layers.
>
> To address this limitation, our work introduces a reconfiguration of the computation pathway. While this reconfiguration does not modify any part of the pretrained forward computation, it fundamentally changes how the point-estimated segment MLP(LN($Z_t$)) is interpreted. Rather than being an observed function drawn from an unknown prior, it is parameterized as part of a Gaussian transition (Eq.4 and lower part of Figure 2). This reveals a learnable representation medium that is more amenable to uncertainty propagation.
>
> Additionally, previous works model the transition probability $p(X_{t−1} | X_t)$ independently at each timestep (Eq.6), which prevents the model from capturing cross-timestep correlations and hinders calibrated uncertainty estimation. In contrast, we model the entire sequence of transitions through a reverse-time diffusion process with a unified spatiotemporal transition model (Eq. 7). This unified kernel captures temporal correlations, improves generalization across timesteps, and yields substantially better calibrated uncertainty (Fig. 1).

---

> ### Author Response · Authors · 2025-11-25
> **Response to Reviewer UFSu (Part 3/3)**
>
> > Is there any benefits that the GPs give you in terms of disentangling aleatoric and epistemic uncertainty? Maybe you could launch a simple experiment, but this has lower priority compared to the other experiments above.
>
> One particular benefit is that under GP, we can have closed-form decomposition of total uncertainty into epistemic and aleatoric uncertainty.
>
> Specifically, for a GP, the predictive distribution is $p(z' \mid x', X, Z)=\mathcal{N}(\mu, \Sigma)$, with total predictive variance $\Sigma = k(x', x')  - k(x', X)\,(K(X, X) + \sigma_{n}^2 I)^{-1} k(X, x')$. The total uncertainty can be achieved by computing entropy of the total predictive variance, which can be further decomposed into aleatoric uncertainty (based on $\sigma_n$) and epistemic uncertainty. For instance, assuming $z = g(x) + \epsilon$ with $\epsilon \sim \mathbb{N}(0, \sigma_n^2)$, from the law of total entropy we have:
> $H[z'] = \mathbb{E}_{g}[H[z' \mid g(x')]] + I(z'; g(x'))$, where:
> - The first term is aleatoric uncertainty, $\mathbb{E}_{g}[H[z' \mid g(x')]] = \tfrac{1}{2}\log(2\pi \epsilon\sigma_n^2)$
> - The second term is epistemic uncertainty,   $I(z'; g(x')) = \tfrac{1}{2}\log(\Sigma)- \tfrac{1}{2}\log(\sigma_{n}^2)$
>
> Having said that, this closed-form expression exists only for single GP. For multiple stacked GPs, we need to approximate this decomposition using variational inference. This allows us to leverage the rich body of literature on variational GP or variational deep GP. In the current scope, our paper only focuses on accessing or calibrating the total uncertainty (via sampling from the predictive distribution). This can be enabled using uncertainty propagation via the learned transition probability. In future work, we will generalize the current work to estimate and propagate epistemic uncertainty which can help better with OOD detection tasks.

---

### Official Review · Reviewer_8shF · 2025-10-22

**Soundness:** 1
**Presentation:** 2
**Contribution:** 2
**Rating:** 4
**Confidence:** 3

**Summary:**

This paper proposes DIRECTOR, a framework for UQ in pre-trained transformer models. The key idea is to interpret transformer blocks as probabilistic transitions between latent representations, allowing the network to be modeled as a reverse diffusion process. Experiments on vision and language tasks (e.g., CIFAR-10/100, IMDB, CoLA) show that DIRECTOR results on calibration and out-of-distribution robustness compared to standard deterministic transformers and prior UQ methods.

**Strengths:**

1. The paper addresses an important and timely topic in modern machine learning.

2. The experimental evaluation is thorough in the sense that it includes comparisons against numerous baselines across multiple datasets.

**Weaknesses:**

1. The paper treats all forms of uncertainty as a single quantity, without distinguishing between aleatoric and epistemic components. If my understanding is correct, the DIRECTOR framework estimates only the total uncertainty.

2. DIRECTOR requires training a separate diffusion model on top of a pre-trained transformer. This introduces additional computational cost for uncertainty estimation, whereas several existing approaches can extract uncertainty directly from the predictive model without such retraining.

3. It is unclear whether the reported improvements are statistically significant. The paper would benefit from significance testing or error bars to support the claimed performance gains.

4. Minor Issues:
   - The definition of \( U_t \) is missing in the main text.
   - The acronym DIRECTOR feels somewhat forced and unintuitive.
   - The bold formatting at the beginning of paragraphs is inconsistent and at times unnecessary.
   - The paper would benefit from additional background on prior methods, such as the Gaussian Process interpretation of transformers and the KEP framework. This material could be included in the appendix.

**Questions:**

1. What exactly is used as the estimate of uncertainty? It is unclear to me how the paper defines or computes the uncertainty measure.

2. When training the diffusion model, could you incorporate existing methods that already estimate uncertainty in diffusion models [1,2]?

3. What is the statistical significance of the results reported in Tables 1–5?

4. Have you evaluated your method for hallucination detection?

[1] Berry, Lucas, Axel Brando, and David Meger. "Shedding light on large generative networks: Estimating epistemic uncertainty in diffusion models." The 40th Conference on Uncertainty in Artificial Intelligence. 2024.

[2] Berry, Lucas, et al. "Seeing the Unseen: How EMoE Unveils Bias in Text-to-Image Diffusion Models." arXiv preprint arXiv:2505.13273 (2025).

---

> ### Author Response · Authors · 2025-11-25
> **Response to Reviewer 8shF (Part 1/3)**
>
> We would like to thank the reviewer for the insightful feedbacks and we will address the remaining concerns as follows:
>
> > The paper treats all forms of uncertainty as a single quantity, without distinguishing between aleatoric and epistemic components. If my understanding is correct, the DIRECTOR framework estimates only the total uncertainty.
>
> We would like to clarify that our work does not aim to decompose uncertainty into aleatoric and epistemic components. Instead, our focus is on achieving calibrated uncertainty estimates, i.e., ensuring that the model’s predictive confidence aligns with true probability, as defined by the notion of perfect calibration [1]. Since our objective is calibrated predictive uncertainty rather than uncertainty decomposition, we do not view the absence of aleatoric–epistemic separation as a limitation of the proposed framework.
>
> [1] Chuan Guo, Geoff Pleiss, Yu Sun, and Kilian Q Weinberger. On calibration of modern neural
> networks. In International conference on machine learning, pp. 1321–1330. PMLR, 2017.
>
> > DIRECTOR requires training a separate diffusion model on top of a pre-trained transformer. This introduces additional computational cost for uncertainty estimation, whereas several existing approaches can extract uncertainty directly from the predictive model without such retraining.
>
> We agree that methods such as Temperature Scaling (TS) and Monte Carlo Dropout (MCD) can extract uncertainty directly from a pretrained model without additional training. However, this convenience comes at the cost of significantly weaker calibration performance. As shown in Table 2, DIRECTOR consistently outperforms these simpler approaches across all evaluated metrics.
>
> Moreover, while DIRECTOR requires training the unified transition model, the resulting system remains computationally efficient at inference time. The transition model is also lightweight which contains only 2.7M parameters compared to 6.24M in the ViT backbone for vision tasks, and 2.59M compared to 3.38M in the text transformer for NLP tasks (lines 296–297). Thus, the additional training cost yields substantial calibration benefits while keeping memory usage low during deployment.

---

> ### Author Response · Authors · 2025-11-25
> **Response to Reviewer 8shF (Part 2/3)**
>
> > It is unclear whether the reported improvements are statistically significant. The paper would benefit from significance testing or error bars to support the claimed performance gains.
> > What is the statistical significance of the results reported in Tables 1–5?
>
> We would like to clarify that we already report error bars in all tables in both the main text and the appendix, providing a measure of variability for all evaluated methods. Regarding significance testing, we show results for the CV task (CIFAR-10) and the NLP task (CoLA) from Table 1. For CoLA, we additionally perform an extensive evaluation with 40 runs and compute paired t-tests across metrics. In the table below, **most entries have p-value < 0.05**, which indicates that **the improvement is significant**.
>
> |**Dataset**|**Method**|**ACC/MCC**||**AURC**||**AUROC**||**FPR95**||**ECE**||**NLL**||**Brier**||
> |-|-|-|-|-|-|-|-|-|-|-|-|-|-|-|-|
> |||Perf.|p-value|Perf.|p-value|Perf.|p-value|Perf.|p-value|Perf.|p-value|Perf.|p-value|Perf.|p-value|
> |**C-10** (DF=4)|ViT|83.84 $\pm$ 0.09||4.05 $\pm$ 0.11||86.42 $\pm$ 0.37||67.13 $\pm$ 1.98||12.51 $\pm$ 0.20||10.91 $\pm$ 0.39||28.03 $\pm$ 0.15||
> ||**Ours**|**85.67 $\pm$ 0.88**|0.0094|**3.10 $\pm$ 0.45**|0.0054|**87.90 $\pm$ 1.04**|0.0255|**61.92 $\pm$ 1.85**|0.0104|**9.67 $\pm$ 0.62**|0.0008|**6.94 $\pm$ 0.52**|0.0001|**23.54 $\pm$ 1.46**|0.0020|
> ||KEP-1/7|84.52 $\pm$ 0.25||3.52 $\pm$ 0.11||87.52 $\pm$ 0.27||65.14 $\pm$ 1.27||10.93 $\pm$ 0.26||8.21 $\pm$ 0.15||25.80 $\pm$ 0.40||
> ||**Ours**|**86.51 $\pm$ 0.57**|0.0051|**2.78 $\pm$ 0.16**|0.0034|**88.28 $\pm$ 0.16**|0.0065|**61.85 $\pm$ 1.93**|0.0149|**8.61 $\pm$ 0.49**|0.0020|**6.11 $\pm$ 0.27**|0.0003|**21.90 $\pm$ 0.82**|0.0020|
> ||KEP-7/7|82.68 $\pm$ 0.12||4.46 $\pm$ 0.05||85.71 $\pm$ 0.34||66.90 $\pm$ 1.78||**6.95 $\pm$ 0.36**||**5.89 $\pm$ 0.11**||25.90 $\pm$ 0.23||
> ||**Ours**|**87.06 $\pm$ 0.18**|0.0000|**2.57 $\pm$ 0.11**|0.0000|**88.60 $\pm$ 0.49**|0.0001|**61.74 $\pm$ 0.52**|0.0023|8.42 $\pm$ 0.15||5.96 $\pm$ 0.18||**21.11 $\pm$ 0.32**|0.0000|
> |**CoLA** (DF=39)|Transformer|29.92 $\pm$ 1.06||20.80 $\pm$ 1.09||64.22 $\pm$ 1.33||90.01 $\pm$ 2.58||26.44 $\pm$ 1.72||19.66 $\pm$ 3.79||55.09 $\pm$ 2.43||
> ||**Director**|**31.97 $\pm$ 2.26**|0.0000|**19.87 $\pm$ 1.46**|0.0004|64.52 $\pm$ 1.44||**89.94 $\pm$ 2.02**|0.8877|**23.33 $\pm$ 4.51**|0.0001|**15.30 $\pm$ 5.62**|0.0001|51.04 $\pm$ 4.80|0.0000|
> ||KEP-1/5|32.40 $\pm$ 2.32||19.94 $\pm$ 1.66||64.42 $\pm$ 2.36||90.21 $\pm$ 2.12||22.45 $\pm$ 4.94||15.24 $\pm$ 5.89||50.27 $\pm$ 4.65||
> ||**Director**|**31.41 $\pm$ 2.05**|0.0429|**18.91 $\pm$ 1.90**|0.0098|**65.45 $\pm$ 1.72**|0.0269|**89.58** $\pm$ 1.61|0.1484|**17.47 $\pm$ 8.87**|0.0027|**11.87 $\pm$ 5.96**|0.0128|**46.62 $\pm$ 6.91**|0.0069|
> ||KEP-5/5|31.36 $\pm$ 2.73||19.80 $\pm$ 1.92||64.66 $\pm$ 0.98||90.26 $\pm$ 1.15||19.87 $\pm$ 5.69||14.74 $\pm$ 8.28||48.69 $\pm$ 5.57||
> ||**Director**|**31.53 $\pm$ 1.95**|0.7545|**18.58 $\pm$ 2.49**|0.0121|**65.33 $\pm$ 1.93**|0.0361|**88.70 $\pm$ 1.63**|0.0000|**15.74 $\pm$ 8.57**|0.0139|**10.80 $\pm$ 5.91**|0.0174|**44.76 $\pm$ 6.98**|0.0076|
>
> > The definition of ( U_t ) is missing in the main text.
>
> We would like to clarify that the definition of $U_t$ is explicitly provided in the main text. In particular, $U_t$ is defined as the input to the t-th MHSA block, as stated on line 161.
>
> > The bold formatting at the beginning of paragraphs is inconsistent and at times unnecessary.
>
> We appreciate the reviewer’s observation regarding inconsistent bold formatting at the beginning of paragraphs. We will revise the manuscript to ensure that the formatting is applied consistently and only where appropriate.

---

> ### Author Response · Authors · 2025-11-25
> **Response to Reviewer 8shF (Part 3/3)**
>
> > The paper would benefit from additional background on prior methods, such as the Gaussian Process interpretation of transformers and the KEP framework. This material could be included in the appendix.
>
> We would like to clarify that additional background on previous approaches—including the Gaussian Process reinterpretation and the KEP-SVGP framework—was already provided in Appendix A.2.
>
> > What exactly is used as the estimate of uncertainty? It is unclear to me how the paper defines or computes the uncertainty measure.
>
> We would like to clarify that the uncertainty estimates in our work are evaluated through standard calibration metrics, including Negative Log-Likelihood (NLL), Expected Calibration Error (ECE), and Brier Score.
>
> > When training the diffusion model, could you incorporate existing methods that already estimate uncertainty in diffusion models [1,2]?
>
> We thank the reviewer for highlighting relevant methods for uncertainty estimation in diffusion models. These techniques are indeed interesting and could potentially enhance our framework. However, our primary objective in this work is to improve uncertainty calibration for pretrained transformer models by introducing a unified diffusion-based reconfiguration. While uncertainty–estimation methods tailored specifically for diffusion models could, in principle, be incorporated into our unified transition model, doing so would extend beyond the scope of the present study. We view this integration as a promising direction for future work.
>
>
>
> > Have you evaluated your method for hallucination detection?
>
> We would like to clarify that this work focuses exclusively on prediction tasks, following the same evaluation setting as prior methods such as KEP-SVGP [2]. Since hallucination detection pertains to generative modeling, it falls outside the scope of the current study. Nonetheless, extending our framework to generative tasks is an interesting direction that we will explore in future work.
>
>
> [2] Yingyi Chen, Qinghua Tao, Francesco Tonin, and Johan Suykens. Self-attention through kernel-eigen pair sparse variational gaussian processes. In Forty-first International Conference on Machine
> Learning, 2024b.

---

### Official Review · Reviewer_sk5x · 2025-11-01

**Soundness:** 3
**Presentation:** 3
**Contribution:** 2
**Rating:** 4
**Confidence:** 2

**Summary:**

The paper introduces **DIRECTOR**, a method that reinterprets Transformer architectures as **reverse-time diffusion processes** in feature space. By reconfiguring each block to end with a Multi-Head Self-Attention (MHSA) module, the model treats inter-block representations as Gaussian transitions. A unified spatiotemporal kernel then models uncertainty propagation across layers.

The approach is formalized through a **variational upper bound** on the likelihood, yielding a practical objective that combines KL-matching between per-step transitions and a learnable diffusion model, plus a performance-aware regularizer.

Experiments on CIFAR-10, CIFAR-10-C, and CoLA demonstrate improvements in calibration (ECE, NLL), robustness, and OOD detection while maintaining accuracy. The contribution aims to make uncertainty quantification in Transformers more coherent and computationally efficient compared to per-block Gaussian process reparameterizations.

**Strengths:**

- **Elegant conceptual reformulation:** The idea of treating Transformer layers as a unified stochastic diffusion process is creative and intuitively appealing.
- **Mathematically consistent derivation:** The variational objective is well explained and easy to reproduce.
- **Trustworthiness impact:** DIRECTOR provides an interpretable probabilistic mechanism for uncertainty propagation, aligning with responsible AI principles.
- **Consistent empirical results:** Improvements in calibration and robustness are observed across different datasets.
- **Potential for extensibility:** The framework could inspire future work on probabilistic Transformers and diffusion-based calibration.

**Weaknesses:**

1. **Limited novelty:** The work combines existing paradigms (diffusion models, GP interpretations, variational inference) without introducing fundamentally new theory.
2. **Empirical scope:** Experiments are limited to moderate-sized benchmarks; scalability to large models remains untested.
3. **Simplified covariance modeling:** The diagonal and Cholesky-like approximations likely underestimate structured uncertainty.
4. **No runtime comparison:** Computational overhead relative to KEP, SWAG, or ensembles is not reported.
5. **Lack of theoretical guarantees:** The paper does not formally explain why the unified diffusion kernel improves calibration.
6. **Ablation limitations:** The sensitivity of results to loss weight choices and kernel design is not fully explored.

**Questions:**

1. How is the unified kernel \(q_\theta(X_{t-1}|X_t)\) parameterized? Are its parameters shared across time steps or conditioned via embeddings?
2. Have you tested low-rank or structured covariance forms, and if so, how do they affect calibration and runtime?
3. What is the computational and memory overhead of DIRECTOR compared to KEP-all, KEP-last, and deep ensembles?
4. Could you report comparisons with simpler calibration baselines such as temperature scaling or MC dropout?
5. Is there a theoretical link between diffusion-step consistency and improved uncertainty calibration that could explain the observed robustness gains?
6. How might the approach generalize to large models like ViT-B or language models (e.g., GPT architectures)?

---

> ### Author Response · Authors · 2025-11-25
> **Response to Reviewer sk5x (Part 1/2)**
>
> We would like to thank the Reviewer for insightful feedbacks and we will address the remaining concerns as follows:
>
> > Limited novelty: The work combines existing paradigms (diffusion models, GP interpretations, variational inference) without introducing fundamentally new theory.
> > Lack of theoretical guarantees: The paper does not formally explain why the unified diffusion kernel improves calibration.
> > Is there a theoretical link between diffusion-step consistency and improved uncertainty calibration that could explain the observed robustness gains?
>
> We agree that additional theoretical guarantees about link between diffusion-step consistency and improved uncertainty calibration would further strengthen the contribution of this work. However, we note that prior approaches in this area similarly lack formal guarantees for improved uncertainty quantification and predominantly rely on empirical evidence. Developing such theoretical foundations is nontrivial and falls beyond the scope of our current study, but we view it as an important and promising direction for future research.
>
> > Empirical scope: Experiments are limited to moderate-sized benchmarks; scalability to large models remains untested.
> > How might the approach generalize to large models like ViT-B or language models (e.g., GPT architectures)?
>
> We would like to clarify that our experimental setup closely follows the configurations used in the most recent prior work, KEP-SVGP [1], to ensure a fair comparison. More importantly, we emphasize that our method also scales effectively to larger architectures as reported in Table 6 of Appendix A.6.1. Specifically, DIRECTOR’s reconfigured ViT-B-16 achieves substantially better uncertainty calibration with an ECE of 1.611 compared to 18.021 of the original ViT-B-16 (86M parameters). DIRECTOR also maintains competitive predictive performance (97.17% vs. 97.88%) while using significantly fewer parameters than original model (50M vs. 86M).
>
> [1] Yingyi Chen, Qinghua Tao, Francesco Tonin, and Johan Suykens. Self-attention through kernel-eigen pair sparse variational gaussian processes. In Forty-first International Conference on Machine
> Learning, 2024b.
>
> > Simplified covariance modeling: The diagonal and Cholesky-like approximations likely underestimate structured uncertainty.
> >Have you tested low-rank or structured covariance forms, and if so, how do they affect calibration and runtime?
>
> We would like to clarify that we approximate pretrained feature distribution's covariance $\sigma_t(X_t)$ using a Cholesky-like factor used in KEP-SVGP [1] (as we choose KEP-SVGP as pretrained model to match). Moreover, we adopt a diagonal structure for the parameterized covariance $\sigma_\theta(X_t, t)$, which is standard practice in diffusion models [2]. To assess whether this simplification sacrifices calibration quality, we conducted a small-scale experiment comparing the diagonal parameterization with a richer covariance structure.
>
> **Expreriment results on varying covariance structures with ViT as baseline**
> |**Cov. Struct.**|**ACC**|**AURC**|**AUROC**|**FPR95**|**ECE**|**NLL**|**Brier**|**Training time (hours)**|
> |-|-|-|-|-|-|-|-|-|
> |Diagonal|86.38|2.70|88.80|61.16|9.00|6.34|22.21|1.75|
> |Full (dimension-wise)|81.55|4.95|85.48|68.08|10.70|7.28|28.88|2.5|
> |Baseline|83.90|4.02|86.71|65.16|12.57|10.94|27.80||
>
> **Expreriment results on varying covariance structures with KEP-1/7 as baseline**
> |**Cov. Struct.**|**ACC**|**AURC**|**AUROC**|**FPR95**|**ECE**|**NLL**|**Brier**|Training time (hours)|
> |-|-|-|-|-|-|-|-|-|
> |Diagonal|86.22|2.83|88.35|61.90|8.85|6.21|22.23|1.75|
> |Full (dimension-wise)|83.8|3.81|87.26|62.10|11.16|8.12|26.34|2.5|
> |Baseline|84.61|3.52|87.44|65.04|10.86|8.17|25.59||
>
> The results indicate that the diagonal approximation not only achieves faster runtime but also provides superior calibration performance compared to its more complex counterparts. These findings support the efficiency and effectiveness of the diagonal form in our probabilistic reconfiguration.
>
> [2] Ho, Jonathan, Ajay Jain, and Pieter Abbeel. "Denoising diffusion probabilistic models." Advances in neural information processing systems 33 (2020): 6840-6851.

---

> ### Author Response · Authors · 2025-11-25
> **Response to Reviewer sk5x (Part 2/2)**
>
> > No runtime comparison: Computational overhead relative to KEP, SWAG, or ensembles is not reported.
> > What is the computational and memory overhead of DIRECTOR compared to KEP-all, KEP-last, and deep ensembles?
>
> Thank you for the suggestions. We would like to provide additional evaluation of the computational and memory overhead below.
>
> |Method|Inference time (s)|Inference memory (GB)|
> |-|-|-|
> |KEP-7/7|2.1038|0.3348|
> |KEP-1/7|1.5129|0.2916|
> |SGPA|4.5169|0.5768|
> |ViT|1.3173|0.1509|
> |**Ours**|1.6756|0.1040|
>
> As shown in the table, our method achieves substantially better memory efficiency than all baselines, while maintaining an inference time comparable to KEP-1/7.
>
> >Ablation limitations: The sensitivity of results to loss weight choices and kernel design is not fully explored.
>
> We would like to clarify that the effect of varying loss weights is already examined in Appendix A.6.5. Regarding kernel design, while score networks in diffusion models are commonly implemented using MLPs, our experiments show that a DiT-based architecture [4] consistently yields the best performance. The corresponding comparison table is provided below, and we will include it in the appendix of the revised manuscript.
>
> **Expreriment results on varying unified transition models (UTM) with ViT (single run) as baseline**
> |**UTM**|**ACC**|**AURC**|**AUROC**|**FPR95**|**ECE**|**NLL**|**Brier**|
> |-|-|-|-|-|-|-|-|
> |MLP|81.17|5.11|85.47|67.39|1.36|5.52|26.70|
> |DiT|86.18|2.95|87.74|63.82|8.90|6.27|22.62|
> |Baseline|83.90|4.02|86.71|65.16|12.57|10.94|27.80|
>
> **Expreriment results on varying unified transition models (UTM) with KEP-7/7 (single run) as baseline**
> |**UTM**|**ACC**|**AURC**|**AUROC**|**FPR95**|**ECE**|**NLL**|**Brier**|
> |-|-|-|-|-|-|-|-|
> |MLP|83.46|4.16|85.95|65.18|4.99|5.34|24.37|
> |DiT|87.37|2.43|88.99|60.97|8.53|6.07|20.84|
> |Baseline|82.34|4.42|86.45|65.80|5.81|5.59|25.68|
>
> [4] William Peebles and Saining Xie. Scalable diffusion models with transformers. arXiv preprint
> arXiv:2212.09748, 2022
>
> >How is the unified kernel (q_\theta(X_{t-1}|X_t)) parameterized? Are its parameters shared across time steps or conditioned via embeddings?
>
> We would like to clarify that the unified kernel $q_\theta(X_{t-1}|X_t)$ parameterization is described in Section 2.2 (lines 215-226 and Eq.7). Specifically, the kernel is defined as $q_\theta(X_{t-1}|X_t)=N(X_{t-1}|m_\theta(X_t,t),\sigma_\theta(X_t,t))$, where the parameter $\theta$ is shared across time steps. In practice, the unified spatiotemporal transition model is implemented using a single-block DiT architecture [4], as detailed in lines 290–297.
>
> > Could you report comparisons with simpler calibration baselines such as temperature scaling or MC dropout?
>
> We would like to clarify that comparisons with standard calibration baselines—including Temperature Scaling (TS) and MC Dropout (MCD)—are already provided in Table 2 of the main text and in the extended results in Table 8 of the Appendix. Across these evaluations, DIRECTOR consistently outperforms these baselines in both predictive accuracy and uncertainty calibration, demonstrating its advantage over simpler post-hoc or stochastic regularization methods.

---

### Official Review · Reviewer_Lkoa · 2025-11-04

**Soundness:** 2
**Presentation:** 1
**Contribution:** 2
**Rating:** 4
**Confidence:** 3

**Summary:**

The paper proposes DIRECTOR, which reconfigures pretrained transformers into a diffusion-like probabilistic model for uncertainty quantification (UQ). Each transformer block is treated as a Gaussian transition, forming a continuous stochastic process that captures correlations across blocks. The learned diffusion kernel then enables principled uncertainty propagation across the network. Experiments span vision (CIFAR-10/100, CIFAR-10-C) and language (IMDB, CoLA) benchmarks, comparing DIRECTOR with ViT, transformers, and other uncertainty-aware models like KEP-SVGP, MC Dropout, and KFLLA. Results show lower calibration error (ECE, NLL, Brier) and competitive or improved accuracy, with added robustness to distribution shifts and OOD detection

**Strengths:**

+ Better or competitive accuracy and calibration performance across CV and NLP, in-dist, OOD, and corruption datasets.

**Weaknesses:**

- The paper assumes readers already know the emerging approaches with the terms "reparameterize the attention outputs", " probabilistic chain mapping", etc, under-explained. It is hard to get the real problem of these emerging approaches, thus weakening the motivation of the proposed method. The core motivation, "uncertainty doesn’t propagate properly when each block is reparameterized separately", is supported only by a small, hand-picked comparison that could be confounded.
- In Fig. 1, if cross-block correlation is the culprit, the paper should show stronger diagnostics, e.g., measuring inter-block covariance of features/uncertainty, or ablating correlations explicitly, rather than relying on a single "all-blocks vs last-block" contrast.
- Prior work is said to already "recast the pre-trained transformer as a probabilistic chain", but Contribution 1 then "reinterprets" step-wise transformations as a probability path, which sounds overlapping, and the paper doesn’t draw a crisp boundary between prior "probabilistic chain" framing and this paper’s stated reinterpretation.

**Questions:**

See weaknesses.

---

> ### Author Response · Authors · 2025-11-25
> **Response to Reviewer Lkoa**
>
> We would like to thank the Reviewer for insightful feedbacks and we will address the remaining concerns as follows:
>
> > The paper assumes readers already know the emerging approaches with the terms "reparameterize the attention outputs", " probabilistic chain mapping", etc, under-explained. It is hard to get the real problem of these emerging approaches, thus weakening the motivation of the proposed method. The core motivation, "uncertainty doesn’t propagate properly when each block is reparameterized separately", is supported only by a small, hand-picked comparison that could be confounded.
>
>
> We would like to clarify that the emerging paradigms mentioned in the Introduction are already discussed in greater depth in Section 2.1 (lines 153–173). In particular, we provide a general description of GP-based reparameterization of attention outputs, which transforms each MHSA block into a probabilistic transition function, thereby turning the entire transformer into a probabilistic chain mapping. Moreover, we explicitly refer readers to Appendix A.2 (line 163), where we summarize the reparameterization techniques proposed in prior work.
>
> Regarding the concern about the “small, hand-picked comparison,” we emphasize that our core motivation is supported not only by the illustrative visualization in Figure 1, but also by a comprehensive set of experiments. Specifically, we demonstrate that our approach propagates uncertainty more effectively than methods that reparameterize each block independently, across a variety of settings—including in-distribution classification (Tables 1 and 2), OOD detection (Tables 4 and 5), and deep ensembles (Appendix A.6.2). We therefore believe that the motivation for our method is substantiated by robust and wide-ranging empirical evidence.
>
>
> >In Fig. 1, if cross-block correlation is the culprit, the paper should show stronger diagnostics, e.g., measuring inter-block covariance of features/uncertainty, or ablating correlations explicitly, rather than relying on a single "all-blocks vs last-block" contrast.
>
> We thank the Reviewer for this insightful suggestion. Following the recommendation, we provide additional diagnostics to verify that cross-layer feature correlation is indeed the primary factor behind the suboptimal uncertainty quantification observed in prior methods. Specifically, we compare the correlation between features at the first layer ($\mathbf{X}_6$) and and those at deeper layers ($\mathbf{X}_4$, $\mathbf{X}_2$, and $\mathbf{X}_0$ at the last layer) for both DIRECTOR and KEP-All, using a 7-layer model on CIFAR-10. The results are illustrated in Fig. 1c of our updated manuscript and summarized in the table below:
>
> |Method|Corr($X_6, X_4$)|Corr($X_6, X_2$)|Corr($X_6, X_0$)|
> |-|-|-|-|
> |DIRECTOR|0.9751|0.9745|0.9718|
> |KEP-All|0.8440|0.8333|0.6211|
>
> The result indicates that the correlation between the first-layer KEP-SVGP features and later representations drops sharply as it is propagated further towards the solution head. In contrast, our proposed method DIRECTOR manages to correctly preserve this high correlation.
>
>
> >Prior work is said to already "recast the pre-trained transformer as a probabilistic chain", but Contribution 1 then "reinterprets" step-wise transformations as a probability path, which sounds overlapping, and the paper doesn’t draw a crisp boundary between prior "probabilistic chain" framing and this paper’s stated reinterpretation.
>
> Existing "probabilistic chain" framing can be viewed as having separate parameterization at each time step. Such separate reparameterization fails to account for the correlations among feature transformations at different attention blocks that were established during pre-training. Uncertainty thus cannot be propagated consistently across these blocks. This underscores the need for a more robust reparameterizing mechanism that explicitly incorporates such correlations while learning the evolution of representation distributions across the attention blocks (see lines 73-77). This is the focus of our work which intuitively distills probabilistic chains with separately parameterized step-wise transitions into a unified transition model (applied to all steps). Our hypothesis is that such unified parameterization can facilitate more consistent uncertainty propagation (Fig. 1). It has been verified in our experiments where the results show that our method consistently achieved better uncertainty calibration than prior work.

---

### Meta-Review · Area_Chair_rxRW · 2026-01-07

**Summary:**

This paper introduced a posthoc process for pretrained transformer models to turn them into a diffusion-inspired probabilistic framework for uncertainty estimation. The main idea is to treat each Transformer block as a Gaussian transition and model the entire architecture as a unified probability path. The model can more effectively propagate representation uncertainty than existing methods that reparameterize blocks independently. Here is a summary of the Reviewers' concerns:

* Scalability: A primary weakness is the limited scale of the experiments. All evaluations are conducted on relatively small (3M–6M parameter) and shallow (5–7 layer) models. The generalizability to modern, large-scale Transformers (e.g., ViT-L or LLMs) remains unproven.

* Theoretical analysis: While conceptually appealing, the paper lacks a formal theoretical guarantee explaining why a unified diffusion kernel leads to better-calibrated uncertainty compared to other probabilistic interpretations.

* Simplifying Approximations: The reliance on diagonal and Cholesky-like covariance approximations may lead to an underestimation of structured uncertainty, and the sensitivity to hyperparameters (e.g., loss weights) is not fully explored.

* Limited downstream task: Despite using both vision and language data, the evaluation focuses strictly on classification and OOD detection, leaving out generative tasks and hallucination detection,  where UQ is increasingly critical.

**Reviewer Concerns:**

These concerns are still valid:
Inadequate Scalability: The reliance on small-scale datasets and shallow architectures prevents a comprehensive assessment of the method’s efficacy in the context of modern deep learning.

Limited Theoretical Depth: The absence of a rigorous theoretical link between diffusion-step consistency and calibration gains limits the paper’s scientific contribution.

Unresolved Concerns on Generalization: While the rebuttal addressed several technical points, it did not provide evidence that the method scales to the 100M+ parameter range typical of current research, leaving the practical impact of the work in question.

**Reviewer Scores:**

* The reviewer 8shF gave a Rating 4 with Confidence of 3.
* The reviewer sk5x gave a rating of 4 with Confidence of 2
* The reviewer Lkoa gave a rating of 4 with Confidence of 3
* and the reviewer UFSu suggested a Rating: 6 with the lowest confidence (2).

---

### Decision · Program_Chairs · 2026-01-26

Reject